# A Homogenization Approach for Gradient-Dominated Stochastic Optimization

**Jiyuan Tan**[1,*] **Chenyu Xue**[2,*] **Chuwen Zhang**[2] **Qi Deng**[3] **Dongdong Ge**[3] **Yinyu Ye**[1]

[*]Equal Contribution
[1]Department of Management Science and Engineering, Stanford University, Palo Alto, California, USA
[2]Research Institute for Interdisciplinary Sciences, Shanghai University of Finance and Economics, Shanghai, China
[3]Antai College of Economics and Management, Shanghai Jiao Tong University, Shanghai, China

## Abstract

Gradient dominance property is a condition weaker than strong convexity, yet sufficiently ensures global convergence even in non-convex optimization. This property finds wide applications in machine learning, reinforcement learning (RL), and operations management. In this paper, we propose the stochastic homogeneous second-order descent method (SHSODM) for stochastic functions enjoying gradient dominance property based on a recently proposed homogenization approach. Theoretically, we provide its sample complexity analysis, and further present an enhanced result by incorporating variance reduction techniques. Our findings show that SHSODM matches the best-known sample complexity achieved by other second-order methods for gradient-dominated stochastic optimization but without cubic regularization. Empirically, since the homogenization approach only relies on solving extremal eigenvector problem at each iteration instead of Newton-type system, our methods gain the advantage of cheaper computational cost and robustness in ill-conditioned problems. Numerical experiments on several RL tasks demonstrate the better performance of SHSODM compared to other off-the-shelf methods.

## 1 INTRODUCTION

This paper considers the following non-convex optimization problem under the stochastic approximation framework [Robbins and Monro, 1951],

$$\min_{x \in \mathbb{R}^n} F(x) := \mathbb{E}_{\xi \sim \mathcal{D}} \left[ f(x, \xi) \right], \qquad (1)$$

where $\xi$ is the random variable sampled from a distribution $\mathcal{D}$, and $n$ is the dimension of the decision variable $x$. The above framework encompasses a wide range of problems,

ranging from the offline setting, wherein the objective function is minimized over a pre-determined number of samples, to the online setting, where the samples are drawn sequentially from the same distribution.

Since $F(\cdot)$ is non-convex, finding its global optimum is NP-hard in general [Pardalos and Vavasis, 1991, Hillar and Lim, 2013]. Therefore, a practical goal is to find a *stationary point*[1]. From the perspective of sample complexity, such a goal can be achieved using $\mathcal{O}(\epsilon^{-3})$ stochastic gradient and Hessian-vector product. Surprisingly, the result cannot be improved using any stochastic $p$-th order methods for $p \geq 2$ with standard Lipschitz continuity assumptions [Arjevani et al., 2020]. From this point of view, the benefits of second-order information seem limited.

In real life, however, problem (1) usually bears some structures that enable global convergence and fast convergence rate, such as weakly strong convexity [Necoara et al., 2019], error bounds [Luo and Tseng, 1993], and restricted secant inequality [Zhang and Yin, 2013]. Among these conditions, it is shown in Karimi et al. [2016] that the Polyak-Łojasiewicz (PŁ) condition [Polyak et al., 1963] is generally weaker, which can be covered by the *gradient dominance property* studied in this paper. Informally, we say function $F(\cdot)$ is gradient-dominated with parameter $\alpha$ if there exists a constant $C > 0$ such that $F(x) - F(x^*) \leq C\|\nabla F(x)\|^\alpha$, where $x^* \in \arg\min_x F(x)$. When $\alpha = 2$, it reduces to the PŁ condition. This property also enjoys many important applications in different fields, including generalized linear models [Foster et al., 2018], ResNet with linear activation [Hardt and Ma, 2016], operations management [Fatkhullin et al., 2022]. Moreover, Agarwal et al. [2021] show that a weaker version of gradient dominance property with $\alpha = 1$ is also satisfied in policy-based reinforcement learning (RL).

In this paper, we study the sample complexity required by ensuring *global optimality* in non-convex stochastic optimization with the gradient dominance property. Specifically,

---

[1]A stationary point is a point $x$ satisfying $\|\nabla F(x)\| \leq \mathcal{O}(\epsilon)$.

we are interested in the number of queries of stochastic gradient and Hessian along the iterations until reaching a point $x$ that satisfies $\mathbb{E}[F(x)] - F(x^*) \le \epsilon$.

Different from Lipschitz continuity assumptions, gradient dominance property witnesses an improved sample complexity (or iteration complexity) for both stochastic and deterministic second-order methods [Nesterov and Polyak, 2006, Chayti et al., 2023, Masiha et al., 2022] when compared to the first-order methods [Nguyen et al., 2019, Fontaine et al., 2021]. Nevertheless, a common drawback of these second-order methods lies in their dependence on the expensive $\mathcal{O}(n^3)$ computational cost at each iteration to obtain an approximate solution to an inevitable *cubic-regularized subproblem*[2]. Recently, Zhang et al. [2022] develop a novel homogeneous second-order method (HSODM), which only needs to find the leftmost eigenvector of an augmented matrix at each iteration with $\tilde{\mathcal{O}}(n^2)$[3] computational cost [Kuczyński and Woźniakowski, 1992]. They prove that HSODM not only reduces $\mathcal{O}(n^3)$ computational burden required by cubic-regularization, but also achieves the optimal iteration complexity [Carmon et al., 2021].

Hence, a natural question arises:

*Can the homogenization approach be extended to gradient-dominated stochastic optimization, and maintain the state-of-the-art sample complexity?*

In this paper, we give an affirmative answer to the question above. Our contributions are summarized as follows:

1. First, we propose two non-trivial customized strategies to extend HSODM from non-convex optimization to gradient-dominated optimization. The success of HSODM is attributed to its fixed choice of the last diagonal element of the augmented matrix, which can not directly apply to gradient-dominated optimization (we will explain it in Section 3). To tackle this challenge, we develop a perturbation strategy and design a novel parameter-searching strategy to construct the augmented matrix, both of which significantly differs from HSODM, and may be of independent interest to extend HSODM to convex optimization.

2. Second, we propose a variant of HSODM for gradient-dominated stochastic optimization, SHSODM, and analyze its sample complexity for stochastic function enjoying gradient dominance property with $\alpha \in [1, 2]$. When $\alpha \in [1, 3/2)$, we further provide an enhanced result by employing variance reduction techniques. Specifically, the sample complexity of SHSODM can

be improved to $\mathcal{O}\left(\epsilon^{-2/\alpha}\right)$. Our results match the best-known sample complexity in the literature obtained by the stochastic cubic-regularized Newton method (SCRN, Masiha et al. 2022). For clarity, we give the detailed results[4] in Table 1.

3. Finally, SHSODM only requires solving an eigenvalue problem at each iteration, hence overcoming the heavy computational burden of second-order methods. Empirically, we test SHSODM in the context of RL, whose objective function enjoys gradient dominance property with $\alpha = 1$, and compare its performance with SCRN and other standard RL algorithms. The numerical experiments demonstrate that SHSODM is superior to these methods and immune to ill-conditioning.

The rest of the paper is organized in the following manner. In the remainder of the section, we review some related literature. Section 2 gives a formal definition of gradient dominance property considered in this paper, and a brief introduction to HSODM. In Section 3, we propose two nontrivial customized strategies to extend HSODM to gradient-dominated optimization. In Section 4.1, we formally describe our SHSODM and prove its sample complexity, which is improved later in Section 4.2 by adapting the variance reduction technique. Section 5 provides the numerical experiments in RL, and demonstrates the superior performance of SHSODM to SCRN and other widely-used RL algorithms. Finally, Section 6 concludes the paper and presents several future research directions.

## 1.1  RELATED WORK

We review some papers studying gradient dominance property and second-order methods for gradient-dominated optimization in both deterministic and stochastic settings. The more comprehensive literature review including first-order methods is provided in the Appendix.

**Gradient-dominated optimization and its applications.** The gradient dominance property with $\alpha = 2$ (or the PŁ condition) is first introduced in Polyak et al. [1963]. It is strictly weaker than strong convexity which is sufficient to guarantee the global linear convergence rate for the first-order methods. Karimi et al. [2016] further show that the PŁ condition is weaker than most of the global optimality conditions that appeared in the machine learning community. The gradient dominance property is also established locally or globally under some mild assumptions for problems such as phase retrieval [Zhou and Liang, 2017], blind deconvolution [Li et al., 2019], neural network with one hidden layer [Li and Yuan, 2017, Zhou and Liang, 2017], linear residual neural networks [Hardt and Ma, 2016], and

---

[2]In our discussion, we solve linear systems via computing the matrix inverse directly, or the direct method. Although there are some indirect methods for solving linear systems, they either have worse dependence on condition number, or have the same computational cost as our proposed method.

[3]In this paper, we use $\tilde{\mathcal{O}}(\cdot)$ to ignore the logarithmic factors.

---

[4]In Masiha et al. [2022], they do not give the sample complexity of SCRN with variance reduction explicitly when $\alpha \in [1, 3/2)$. However, by the similar technique they use when $\alpha = 1$, we derive the corresponding result and present it here.

Table 1: Sample complexity of different algorithms for gradient-dominated stochastic optimization. The third to sixth columns present the sample complexity, per iteration cost, whether the algorithm needs to solve linear systems at each iteration, and whether it matches the best-known result, respectively. Note that SGD represents stochastic gradient descent method, the prefix "VR" stands for the variance reduction version, and "w.h.p." stands for "with high probability".

| Algorithm | $\alpha$ | Sample Complexity | Per Iteration Cost | Free of Linear System | Best-known |
|---|---|---|---|---|---|
| SGD | | $\mathcal{O}(\epsilon^{-4/\alpha+1})$ | $\mathcal{O}(n)$ | ✓ | ✗ |
| SCRN | | $\mathcal{O}(\epsilon^{-7/(2\alpha)+1})$ | $\mathcal{O}(n^3)$ | ✗ | ✗ |
| VR-SCRN | $[1, 3/2)$ | $\mathcal{O}(\epsilon^{-2/\alpha})$ | $\mathcal{O}(n^3)$ | ✗ | ✓ |
| **SHSODM** | | $\mathcal{O}(\epsilon^{-7/(2\alpha)+1})$ | $\tilde{\mathcal{O}}(n^2)$ | ✓ | ✗ |
| **VR-SHSODM** | | $\mathcal{O}(\epsilon^{-2/\alpha})$ | $\tilde{\mathcal{O}}(n^2)$ | ✓ | ✓ |
| SGD | | $\mathcal{O}(\epsilon^{-5/3})$ | $\mathcal{O}(n)$ | ✓ | ✗ |
| SCRN | $3/2$ | $\mathcal{O}(\epsilon^{-4/3}\log(1/\epsilon))$ | $\mathcal{O}(n^3)$ | ✗ | ✓ |
| **SHSODM** | | $\mathcal{O}(\epsilon^{-4/3}\log(1/\epsilon))$ | $\tilde{\mathcal{O}}(n^2)$ | ✓ | ✓ |
| SGD | | $\mathcal{O}(\epsilon^{-4/\alpha+1})$ | $\mathcal{O}(n)$ | ✓ | ✗ |
| SCRN | $(3/2, 2]$ | $\mathcal{O}(\epsilon^{-2/\alpha}\log\log(1/\epsilon))$ w.h.p. | $\mathcal{O}(n^3)$ | ✗ | ✓ |
| **SHSODM** | | $\mathcal{O}(\epsilon^{-2/\alpha}\log\log(1/\epsilon))$ | $\tilde{\mathcal{O}}(n^2)$ | ✓ | ✓ |

generalized linear model and robust regression [Foster et al., 2018]. Meanwhile, it is worth noting that in policy-based RL, a weak version of gradient dominance property with $\alpha = 1$ holds for some certain classes of policies, such as the Gaussian policy [Yuan et al., 2022].

**Second-order methods.** The original analysis for second-order methods under gradient dominance property appears in Nesterov and Polyak [2006]. They focus on the cubic-regularized Newton method (CRN) for $\alpha \in \{1, 2\}$. When $\alpha = 2$, they show that the algorithm has a superlinear convergence rate. When $\alpha = 1$, they prove that CRN has a two-phase pattern of convergence. The initial phase terminates superlinearly, while the second phase achieves an iteration complexity of $\mathcal{O}(\epsilon^{-1/2})$. Afterward, the result in Zhou et al. [2018] gives a more fine-grained analysis of CRN for some functions satisfying KŁ property (it covers the gradient dominance property except for the case $\alpha = 1$) by partitioning the interval $(1, 2]$ into $(1, 3/2)$, $\{3/2\}$, and $(3/2, 2]$, which enjoy sublinear, linear, and superlinear convergence rate, respectively. When it comes to the stochastic setting, Masiha et al. [2022] obtain the sample complexity of SCRN for gradient-dominated stochastic optimization, which is the best-known result under this setting. Recently, Chayti et al. [2023] consider the finite-sum setting. Nevertheless, all these papers rely on approximate solutions of cubic-regularized sub-problems.

## 2 PRELIMINARIES

In this section, we provide the preliminaries of our paper and introduce the novel second-order method, HSODM. First, We formally define the gradient dominance property.

**Assumption 2.1** (Gradient Dominance). *We say function $F(x)$ has the weak gradient dominance property with $\alpha \in [1, 2]$, if there exist $C_{weak} > 0$ and $\epsilon_{weak} > 0$ such that for all $x \in \mathbb{R}^n$, it holds that*

$$F(x) - F(x^*) \leq C_{weak}\|\nabla F(x)\|^\alpha + \epsilon_{weak} \quad (2)$$

*where $x^* \in \arg\min F(x)$.*

*Furthermore, if $\epsilon_{weak} = 0$, we say function $F(x)$ has the strong gradient dominance property, that is,*

$$F(x) - F(x^*) \leq C_{gd}\|\nabla F(x)\|^\alpha \quad (3)$$

In this paper, we consider the gradient-dominated function with $\alpha \in [1, 2]$. Note again when $\alpha = 2$, (3) is the PŁ condition. If $\alpha = 1$, (2) relates to the gradient dominance property discussed in policy-based RL. We also refer readers to Fatkhullin et al. [2022] for more concrete examples.

**A Brief Overview of HSODM.** We now introduce the framework of HSODM. The key ingredient of HSODM is the homogenized quadratic model (HQM) constructed at each iterate $x_k, k = 1, 2, \ldots K$. At the $k$-th iteration, it builds a gradient-Hessian augmented matrix $A_k$ and solves a homogeneous quadratic optimization problem. Specifically, the following optimization problem is considered:

$$\min_{\|[v;t]\| \leq 1} \begin{bmatrix} v \\ t \end{bmatrix}^T \begin{bmatrix} H_k & g_k \\ g_k^T & -\delta \end{bmatrix} \begin{bmatrix} v \\ t \end{bmatrix}, \quad (4)$$

where $g_k$ is the gradient and $H_k$ is the Hessian. For ease of exposition, we define $A_k = [H_k, g_k; g_k^T, -\delta]$ and let $[v_k; t_k]$ be the optimal solution to the above problem (4). In the next lemma, we characterize the optimal solution $[v_k; t_k]$.

**Lemma 2.1** (Zhang et al., 2022). *Denote by $[v_k; t_k]$ the optimal solution to problem* (4). *We have:*

*(1) There exists a dual variable $\theta_k \geq 0$ such that*

$$\begin{bmatrix} H_k + \theta_k \cdot I & g_k \\ g_k^T & -\delta + \theta_k \end{bmatrix} \succeq 0, \qquad (5)$$

$$\begin{bmatrix} H_k + \theta_k \cdot I & g_k \\ g_k^T & -\delta + \theta_k \end{bmatrix} \begin{bmatrix} v_k \\ t_k \end{bmatrix} = 0, \qquad (6)$$

$$\theta_k \cdot (\|[v_k; t_k]\| - 1) = 0. \qquad (7)$$

*Moreover, $-\theta_k = \lambda_{\min}(A_k)$.*

*(2) If $t_k \neq 0$, then it holds that*

$$g_k^T d_k = \delta - \theta_k, \quad (H_k + \theta_k \cdot I)d_k = -g_k, \qquad (8)$$

*where $d_k = v_k/t_k$.*

*(3) If $t_k = 0$, $-\theta_k$ is the smallest eigenvalue of $H_k$ and $g_k^T v_k = 0$.*

Lemma 2.1 states that, if one sets $\delta \geq 0$ for the non-convex function, the augmented matrix $A_k$ must be negative definite. Meanwhile, the negative optimal dual solution, i.e., $-\theta_k$, is the smallest eigenvalue of $A_k$ and the optimal primal solution $[v_k, t_k]$ is its associated eigenvector. Moreover, when $t_k \neq 0$, the constructed direction $d_k$ is a descent direction by (8). Hence, one can update the next iterate by $x_{k+1} = x_k + \eta_k d_k$, where $\eta_k > 0$ is the stepsize. The convergence analysis of HSODM under the deterministic non-convex setting heavily depends on the fixed choice of $\delta = \Theta(\sqrt{\epsilon})$. However, this strategy for choosing $\delta$ cannot directly apply to gradient-dominated stochastic optimization, which will be discussed in detail in Section 3.

From the perspective of computational complexity, solving (4) is essentially solving an extreme eigenvalue problem with respect to the augmented matrix $A_k$, since it has been shown by (7) that $[v_k; t_k]$ always attains the boundary of the unit ball. In view of this, the subproblems can be solved within the time complexity of $\tilde{\mathcal{O}}(n^2)$ [Kuczyński and Woźniakowski, 1992], enjoying cheaper computation than that in classical CRN [Nesterov and Polyak, 2006] and its stochastic counterpart, where $\mathcal{O}(n^3)$ arithmetic operations are unavoidable. Similar to any second-order methods, HSODM also benefits from Hessian-vector product (HVP), where the Hessian matrix itself is unnecessary to be stored.

**Remark 2.1.** *To solve* (4)*, one can apply some Lanczos-type algorithms [Kuczyński and Woźniakowski, 1992], which only need access to the oracle $A_k[v; t]$ for any given $[v; t]$. To achieve this, one can compute it by*

$$A_k[v; t] = \begin{bmatrix} H_k & g_k \\ g_k^T & -\delta_k \end{bmatrix} \begin{bmatrix} v \\ t \end{bmatrix} = \begin{bmatrix} H_k v + t g_k \\ g_k^T v - t \delta_k \end{bmatrix}.$$

*Hence, only HVP $H_k v$ is needed.*

## 3 CUSTOMIZED STRATEGIES

In this section, we introduce two customized strategies to extend HSODM to the gradient-dominated world. As HSODM

is originally designed for non-convex optimization, it cannot directly apply to gradient-dominated functions, which also envelop convex functions with a bounded set. When the fixed choice rule of $\delta$ is used, we cannot connect the decrease in function value to the gradient. Consequently, HSODM only attains an unsatisfactory convergence rate, and several refinements must be adopted to fit our purpose.

Inspired by the analysis of CRN [Nesterov and Polyak, 2006], we need to ensure that $\lambda_k$ and $\|d_k\|$ have the same order and diminish simultaneously when the algorithm proceeds, where $\lambda_k = \lambda_{\min}(A_k)$. In other words, we need to approximately solve

$$h(\delta_k) = \lambda_k - C_e \|d_k\| = 0, \qquad (9)$$

where $C_e > 0$ is a pre-specified constant. However, this desired relationship cannot be guaranteed by the fixed strategy to choose $\delta$ employed by HSODM. To address this challenge, we propose to adaptively choose $\delta_k$ at each iteration $k$ by a nontrivial line-search procedure.

Before delving into the analysis, we first parameterize the augmented matrix $A_k$ via $\delta_k$. Specifically, we let

$$A_k(\delta_k) = \begin{bmatrix} H_k & g_k \\ g_k^T & -\delta_k \end{bmatrix}.$$

Thus, $\lambda_k$ and $d_k$ can also be seen as functions of $\delta_k$ (recall that $[v_k; t_k]$ is the leftmost eigenvalue of $A_k(\delta_k)$):

$$d_k(\delta_k) := v_k/t_k, \lambda_k(\delta_k) = \lambda_{\min}(A_k(\delta_k)).$$

Therefore, we are now able to adjust $\delta_k$ of $A_k$ to find a better descent direction such that $\lambda_k$ and $\|d_k\|$ have the same order.

---

**Algorithm 1:** Linesearch

**Input:** Current iterate $x_k$, $H_k, g_k$, tolerance $\epsilon_{\text{ls}}, \epsilon_{\text{eig}}$, initial search interval $[\delta_l, \delta_r]$, ratio $C_e$.

Call Algorithm 2 with $g_k, H_k, \epsilon_{\text{eig}}$ and collect $g_k'$;
**for** $j = 1, ..., J_k$ **do**
    Let $\delta_m = (\delta_l + \delta_r)/2$ ;
    Construct $A_k(\delta_k) := [H_k, g_k'; (g_k')^T, \delta_m]$;
    Calculate the leftmost eigenpair $(\lambda_k, [v_k; t_k])$ of $A_k(\delta_m)$;
    Calculate $d_k := v_k/t_k$;
    **if** $C_e \|d_k\| \leq |\lambda_k|$ **then**
        $\delta_l \leftarrow \delta_m$;
    **else**
        $\delta_r \leftarrow \delta_m$;
    **if** $|\delta_r - \delta_l| < \epsilon_{\text{ls}}$ **then**
        **return** $\delta_m, d_k$;
    **else**
        $j \leftarrow j + 1$;

---

In Algorithm 1, we provide the linesearch procedure to adaptively adjust $\delta_k$ at each iteration $k$. In particular, we

use binary search to find an appropriate $\delta_k$. We terminate the procedure if the search interval is sufficiently small, and conclude that the dual variable $\lambda_k$ approximately has the order of $\|d_k\|$. However, the caveat of the above procedure is that a degenerate solution may exist if $g_k$ is orthogonal to the leftmost eigenspace $\mathcal{S}_{\min}$ of $H_k$. In this case, the eigenvector provides no information about the gradient and (9) may not have a solution. Such a cumbersome case is often regarded as a "hard case" in the literature of trust-region methods [Conn et al., 2000]. To overcome this obstacle, we use a random perturbation over $g_k$, through which the perturbed gradient $g_k'$ is no longer orthogonal to the minimal eigenvalue space $\mathcal{S}_{\min}$, i.e., $\mathcal{P}_{\mathcal{S}_{\min}}(g_k') \geq \epsilon_{\mathrm{eig}}$, where $\epsilon_{\mathrm{eig}}$ is the pre-determined tolerance and $\mathcal{P}_{\mathcal{S}_{\min}}(\cdot)$ represents the projection of a given vector onto $\mathcal{S}_{\min}$.

We prove in the Appendix that, with the linesearch strategy after perturbation, an approximate solution to (9) can always be found due to that $\lambda_k$ is continuous over $\delta_k$. Interestingly, such perturbation strategy is "one-shot" if needed, whose details are presented in Algorithm 2. For cases where $\mathcal{S}_{\min}$ consists of multiple eigenvectors, one can simply compute the inner product of $g_k$ and any $v \in \mathcal{S}_{\min}$, and adopt the perturbation if it is not sufficiently bounded away from zero. Besides, it is known that power method also applies in such scenarios, since only the projection is needed [Golub and Van Loan, 2013]. The finite-step termination property of Algorithm 1 is guaranteed by the following theorem.

---

**Algorithm 2:** A Perturbation Strategy

**Input:** Current iterate $x_k$, $H_k$, $g_k$, tolerance $\epsilon_{\mathrm{eig}}$.
Compute the projection of $g$ to the minimal eigenvalue space $\mathcal{P}_{\mathcal{S}_{\min}}(g)$ ;
**if** $\|\mathcal{P}_{\mathcal{S}_{\min}}(g)\| \geqslant \epsilon_{eig}$ **then**
   |   **return** $g$;
**else**
   |   $g' \leftarrow g + \epsilon_{\mathrm{eig}} \cdot \mathcal{P}_{\mathcal{S}_{\min}}(g)/\|\mathcal{P}_{\mathcal{S}_{\min}}(g)\|$;
   |   **return** $g'$;

---

**Theorem 3.1** (Finite-step Termination of Algorithm 1)**.** *Algorithm 1 terminates in $\mathcal{O}(\log(1/\epsilon_{ls}\epsilon_{eig}))$ steps. Furthermore, it produces an estimate $\hat{\delta}_{C_e}$ of $\delta_k$ and the direction $d_k$ such that $|h(\hat{\delta}_{C_e})| \leq \epsilon_{ls}$, where $\epsilon_{ls} > 0$ is the tolerance.*

The above theorem shows that the number of iterations $J_k$ for Algorithm 1 is at most $\mathcal{O}(\log(1/\epsilon_{\mathrm{ls}}\epsilon_{\mathrm{eig}}))$. The extra invoking of Algorithm 2 is only necessary if we find $g_k \perp \mathcal{S}_{\min}$. Since the computational complexity of Algorithm 2 is within $\tilde{\mathcal{O}}(n^2 \epsilon^{-1/4})$ arithmetic operations and consistent with the computational cost of Algorithm 1, we still harbor an advantage of cheap computational cost compared to solving cubic-regularized subproblems.

# 4  SHSODM FOR GRADIENT-DOMINATED STOCHASTIC OPTIMIZATION

In Section 4.1, we give the details of our SHSODM for the gradient-dominated stochastic optimization when $\alpha \in [1, 2]$ and provide its sample complexity analysis. For the case $\alpha \in [1, 3/2)$, we further incorporate the variance reduction techniques into SHSODM and present an improved sample complexity result in Section 4.2.

## 4.1  SHSODM

We begin by outlining the key steps of SHSODM. Firstly, we randomly draw a sample set $S$, and use it to construct the stochastic approximation of the gradient and the Hessian, respectively, by $\hat{g}_k = \nabla_S F(x_k) = \sum_{i=1}^{|S|} \nabla f(x, \xi_i)/|S|$ and $\hat{H}_k = \nabla_S^2 F(x_k) = \sum_{i=1}^{|S|} \nabla^2 f(x, \xi_i)/|S|$. Then, at each iteration $k$, we employ Algorithm 1 and Algorithm 2 (if necessary) to obtain the update direction $d_k$, which must terminate in $\mathcal{O}(\log(1/\epsilon_{\mathrm{ls}}\epsilon_{\mathrm{eig}}))$ iterations. Finally, we choose the stepsize $\eta_k = 1$ for all $k$ and update $x_{k+1} = x_k + d_k$. The details are provided in Algorithm 3.

Before analyzing the sample complexity of SHSODM, we make some assumptions used throughout the paper.

**Assumption 4.1.** *Assume that function $F(\cdot)$ is twice differentiable. The gradient of $f(x, \xi)$ and the Hessian of $F(x)$ are Lipschitz continuous, respectively. That is, there exists $L_g > 0$ and $L_H > 0$ such that $\|\nabla f(x, \xi) - \nabla f(x, \xi)\| \leq L_g\|x - y\|$ and $\|\nabla^2 F(x) - \nabla^2 F(y)\| \leq L_H\|x - y\|$ for all $x, y \in \mathbb{R}^n$ and $\xi$ almost surely.*

**Assumption 4.2** (Masiha et al. 2022)**.** *Assume that for each query point $x \in \mathbb{R}^d$, the stochastic gradient and Hessian are unbiased, and their variance satisfies $\mathbb{E}[\|\nabla F(x) - \nabla f(x, \xi)\|^2] \leq \sigma_g^2$ and $\mathbb{E}[\|\nabla^2 F(x) - \nabla^2 f(x, \xi)\|^{2\alpha}] \leq \sigma_{h,\alpha}^2$, where $\sigma_g > 0$ and $\sigma_{h,\alpha} > 0$ are two constants.*

Assumption 4.1 ensures the Hessian of function $F(\cdot)$ and the gradient of stochastic function $f(x, \xi)$ are both Lipschitz continuous, which is widely used in the optimization literature [Nesterov, 2018, Masiha et al., 2022, Chayti et al., 2023]. Assumption 4.2 guarantees the stochastic gradient and Hessian are unbiased and have bounded variances, which is standard in stochastic optimization and also used by Masiha et al. [2022].

**Remark 4.1.** *When analyzing SGD under the non-convex setting, Khaled and Richtárik [2022] propose the expected smoothness assumption, which is weaker than Assumption 4.2. However, it is difficult to analyze the behavior of second-order algorithms under this assumption, since it cannot control the term $\mathcal{O}(\mathbb{E}[\|g_k - \hat{g}_k\|^2]^{\alpha/2} + \mathbb{E}[\|H_k - \hat{H}_k\|^{2\alpha}])$ emerging in the analysis, which critically influences the sample complexity. Hence, We leave the relaxation of Assump-*

---

**Algorithm 3:** SHSODM

---

**Input:** Total number of iterations $K$, sample size
$\quad\quad n_g, n_H$, tolerance $\epsilon_{\text{ls}}, \epsilon_{\text{eig}}$, lower bound $\delta_l$ and
$\quad\quad$ upper bound $\delta_r$ of the linesearch procedure

**for** $k \leftarrow 1$ **to** $K$ **do**
$\quad$ Draw samples with $|S_k^g| = n_g$ and $|S_k^H| = n_H$
$\quad$ Construct the empirical estimators $\hat{g}_k$ and $\hat{H}_k$
$\quad$ Call Algorithm 1 with $(\hat{g}_k, \hat{H}_k, \epsilon_{\text{ls}}, \epsilon_{\text{eig}}, \delta_l, \delta_r)$ to
$\quad\quad$ obtain $\delta_k$ and $d_k$;
$\quad$ Update current point $x_{k+1} = x_k + d_k$.;

**return** $x_K$;

---

In the next theorem, we give the sample complexity of SHSODM. It partitions the interval $[1, 2]$ into three non-overlapping subsets, and provides respectively the corresponding sample complexity. When $\alpha = 1$, SHSODM achieves the worst sample complexity of $\mathcal{O}(\epsilon^{-2.5})$. While when $\alpha = 2$, it achieves the best sample complexity of $\tilde{\mathcal{O}}(\epsilon^{-2.5})$. This result matches the one of SCRN [Masiha et al., 2022]. However, our SHSODM does not need to solve linear systems, hence its per-iteration cost is less than SCRN's, which is also validated by Section 5. We also remark that, theoretically, we only need the access to stochastic gradient and HVP, instead of stochastic Hessian. We present the remaining theorems below in the form of sampling Hessian only for the comparison with SCRN.

**Theorem 4.1** (Sample Complexity of SHSODM). *Suppose that function $F(\cdot)$ satisfies Assumption 4.1, Assumption 4.2 and Assumption 2.1 with equation (3) for some $\alpha \in [1, 2]$. Given tolerance $\epsilon$ and let $n_g = \mathcal{O}(\epsilon^{-2/\alpha})$, $n_H = \mathcal{O}(\epsilon^{-1/\alpha})$, $\epsilon_{eig} = \mathcal{O}(\epsilon^{1/\alpha})$, and $\epsilon_{ls} = \mathcal{O}(\epsilon^{1/\alpha})$. Then Algorithm 3 outputs a solution $x_K$ such that $\mathbb{E}[F(x_K) - F(x^*)] \le \epsilon$ after $K$ iterations, where*

*(1) If $\alpha \in [1, 3/2)$, then $K = \mathcal{O}(\epsilon^{-3/(2\alpha)+1)})$ with a sample complexity of $\mathcal{O}(\epsilon^{-7/(2\alpha)+1})$.*

*(2) If $\alpha = 3/2$, then $K = \mathcal{O}(\log(1/\epsilon))$ with a sample complexity of $O\left(\log(1/\epsilon)\epsilon^{-2/\alpha}\right)$.*

*(3) If $\alpha \in (3/2, 2]$, then $K = \mathcal{O}(\log\log(1/\epsilon))$ with a sample complexity of $O\left(\log\log(1/\epsilon)\epsilon^{-2/\alpha}\right)$.*

In the context of policy-based RL, Assumption 2.1 holds with equation (2) for $\alpha = 1$. The following corollary emphasizes the sample complexity of SHSODM under this specific setting, which is used in our numerical experiments.

**Corollary 4.1** (Informal, Sample Complexity for Policy-Based RL). *Under policy-based RL, Assumption 4.1 and Assumption 4.2 hold. Moreover, Assumption 2.1 holds with equation (2) for $\alpha = 1$. Then, Algorithm 3 outputs a solution $x_K$ such that $\mathbb{E}[F(x_K) - F(x^*)] < \epsilon + \epsilon_{weak}$ with a sample complexity of $\mathcal{O}(\epsilon^{-2.5})$.*

As a byproduct, we also study the performance of the deterministic counterpart of SHSODM for the gradient-dominated function. The next corollary shows it matches the convergence rate proved for CRN [Nesterov and Polyak, 2006], and still avoids solving the linear system.

**Corollary 4.2** (Deterministic Setting). *Under the gradient-dominated deterministic setting, the iteration complexity of the deterministic counterpart of SHSODM is $\mathcal{O}(\epsilon^{-3/(2\alpha)+1})$ when $\alpha \in [1, 3/2)$, $O(\log(1/\epsilon))$ when $\alpha = 3/2$, and $O(\log\log(1/\epsilon))$ when $\alpha \in (3/2, 2]$.*

## 4.2 SHSODM WITH VARIANCE REDUCTION

In the previous analysis, we employ the same batch size at each iteration. In this subsection, we enhance the sample complexity of SHSODM when $\alpha \in [1, 3/2)$ via time-varying batch size and variance reduction technique proposed in Fang et al. [2018]. In particular, we let $K_C$ be the period length and $S_k$ be the sample set to estimate gradient at iteration $k$. Then, we construct different empirical estimators of the gradient and the Hessian based on whether $k$ is a multiple of $K_C$. In Algorithm 4, we present the explicit forms of the newly introduced estimators, and give the details of SHSODM with variance reduction technique, or VR-SHSODM mentioned in Table 1.

The next theorem specifies the choice of time-varying batch size and shows the sample complexity of VR-SHSODM can be further improved to $O\left(\epsilon^{-2/\alpha}\right)$ when $\alpha \in [1, 3/2)$. This also applies to weak gradient dominance property with $\alpha \in [1, 3/2)$. When $\alpha = 1$, we enhance the sample complexity from $\mathcal{O}(\epsilon^{-2.5})$ in the last subsection to $\mathcal{O}(\epsilon^{-2})$, which improves upon the best-known sample complexity of SGD (or stochastic policy gradient method for RL) and matches the result of SCRN.

**Theorem 4.2** (Sample Complexity of VR-SHSODM). *Under the same assumptions of Theorem 4.1, if $\alpha \in [1, 3/2)$, $K_C = \mathcal{O}(K)$, and*

$$n_{k,g} = \begin{cases} \mathcal{O}(k^{4/(3-2\alpha)}) & k \bmod K_C = 0 \\ O\left(\|d_k\|^2 K_C(\lfloor k/K_C \rfloor K_C)^{4/(3-2\alpha)}\right) & k \bmod K_C \neq 0 \end{cases}$$

*Then, Algorithm 4 outputs a solution $x_K$ such that $\mathbb{E}[F(x_K) - F(x^*)] \le \epsilon$ in $K = \mathcal{O}(\epsilon^{-3/(2\alpha)+1})$ iteration with sample complexity $O\left(\epsilon^{-2/\alpha}\right)$.*

## 5 NUMERICAL EXPERIMENTS

In this section, we evaluate the empirical performance of our SHSODM in the context of RL, which serves as a standard scenario for gradient-dominated stochastic optimization [Masiha et al., 2022]. In RL, the interaction between the agent and the environment is often described by an infinite-horizon, discounted Markov decision process (MDP), and the agent aims to maximize the discounted cumulative reward. Due to the page limit, we leave the brief introduction

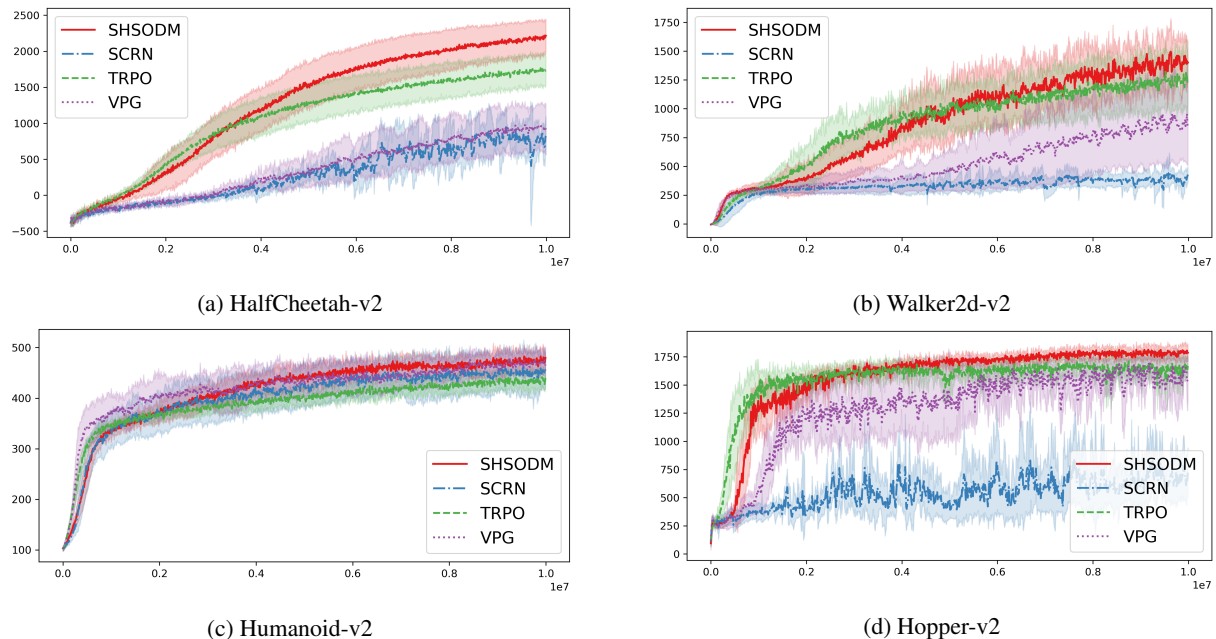

(a) HalfCheetah-v2

(b) Walker2d-v2

(c) Humanoid-v2

(d) Hopper-v2

Figure 1: **The $x$-axis and $y$-axis represents respectively system probes and the average return.** The solid curves depict the mean values of five independent simulations, while the shaded areas correspond to the standard deviation.

---

**Algorithm 4:** SHSODM with Variance Reduction

**Input:** maximum iteration $K$, parameters $K_C$, $\epsilon_{ls}$, $\epsilon_{eig}$

**for** $k \leftarrow 1$ **to** $K$ **do**

    Draw samples with $|S_k| = n_{k,g}$;

    **if** $k \bmod K_C = 0$ **then**

        $v_k \leftarrow \nabla_{S_k} f(x_k)$;

        $H_k \leftarrow \nabla^2_{S_k} f(x_k)$;

    **else**

        $v_k \leftarrow \nabla_{S_k} f(x_k) - \nabla_{S_k} f(x_{k-1}) + v_{k-1}$ ;

        $H_k \leftarrow \nabla^2_{S_k} f(x_k) - \nabla^2_{S_k} f(x_{k-1}) + H_{k-1}$ ;

    Call Algorithm 1 with $\hat{g}_k, \hat{H}_k, \epsilon_{ls}, \epsilon_{eig}, \delta_l, \delta_r$ to get $\delta_k, d_k$;

    Update current point $x_{k+1} = x_k + d_k$;

**return** $x_K$;

---

of MDP in the Appendix. It can be shown under some standard assumptions that the objective function of MDP has the gradient dominance property with $\alpha = 1$.

We compare our SHSODM with different algorithms, including SCRN [Masiha et al., 2022], TRPO [Schulman et al., 2015] and VPG [Williams, 1992]. In the Appendix, we further provide the experiments that compare SHSODM with PPO [Schulman et al., 2017]. As mentioned before, SCRN is a second-order method that harbors the best-known sample complexity for gradient-dominated stochastic optimization. TRPO, and its variants of PPO, are among the most important workhorses behind deep RL and enjoy empirical success. Practically speaking, TRPO can be seen as

a "second-order" method, since it uses the second-order information of the constraints to update the policy. The first-order method VPG is included to serve as a benchmark and validate our theoretical analysis, which is a common practice in the literature [Tripuraneni et al., 2018, Kohler and Lucchi, 2017, Zhou et al., 2018]. We implement SHSODM by using garage library [garage contributors, 2019] written with PyTorch [Paszke et al., 2019], which also provides the implementation of TRPO and VPG. For SCRN, we use the open-source code offered by Masiha et al. [2022].

In our experiments, we test several representative robotic locomotion experiments using the MuJoCo[5] simulator in Gym[6]. The task of each experiment is to simulate a robot to achieve the highest return through smooth and safe movements. Specifically, we consider 4 control tasks with continuous action space, including `HalfCheetah-v2`, `Walker2d-v2`, `Humanoid-v2`, and `Hopper-v2`. For each task, we employ a Gaussian multi-layer perceptron (MLP) policy whose mean and variance are parameterized by an MLP with 2 hidden layers of 64 neurons and `tanh` activation function. We let the batch size be $10^4$, and the number of epoch be $10^3$, leading to a total of $10^7$ time step. To ensure a fair comparison, we adapt the same network architecture for all algorithms.

When computing the discounted cumulative reward, a base-

---

[5]`https://www.mujoco.org`.

[6]Gym is an open source Python library for developing and comparing RL algorithms; see `https://github.com/openai/gym`

| Environment | SHSODM | SCRN | TRPO | VPG |
|---|---|---|---|---|
| HalfCheetah-v2 | **2259 ± 217** | 1115 ± 230 | 1822 ± 214 | 1078 ± 319 |
| Walker2d-v2 | **1630 ± 224** | 543 ± 164 | 1389 ± 265 | 1024 ± 448 |
| Humanoid-v2 | **498 ± 20** | 484 ± 43 | 458 ± 20 | 495 ± 28 |
| Hopper-v2 | **1856 ± 74** | 1473 ± 318 | 1752 ± 86 | 1779 ± 67 |

Table 2: Max average return $\pm$ standard deviation over 5 trials of $10^7$ system probes. Maximum value for each task is bolded.

line term is subtracted in order to reduce the variance. It is noteworthy that the resulting gradient estimator is still unbiased. For all methods, we train a linear feature baseline, which has been implemented in garage. For the hyperparameters of each algorithm, we emphasize that garage's implementations of TRPO and VPG are robust, and tuning parameters does not lead to a significant change of the performance. Hence, we use their default parameters. For SCRN, we use grid search to find the best hyperparameter combination for each environment. For SHSODM, we choose the trust region radius $r \in \{0.05, 0.08\}$. Although our theory always sets it to be 1 (see the constraint in (4)), we make it a hyperparameter in the implementation. To conduct the experiments, we utilize a Linux server with Intel(R) Xeon(R) CPU E5-2680 v4 CPU operating at 2.40GHz and 128 GB of memory, and NVIDIA Tesla V100 GPU.

Following the existing literature [Masiha et al., 2022, Huang et al., 2020], we use the system probes, i.e., the number of sampled state-action pairs, as the measure of sample complexity instead of the number of trajectories due to that the different trajectories may have varying lengths. For each task, we run each algorithm with a total of $10^7$ system probes, employing 5 different random seeds for both network initialization and the Gym simulator.

**Overall Comparisons.** Figure 1 presents the training curves over the aforementioned environments in MuJoCo. It is clear that our SHSODM outperforms SCRN and VPG in all tested environments. When compared with TRPO, although SHSODM grows slowly at the beginning, it achieves the best final performance with an obvious margin. It can be also observed that SCRN is less robust and even inferior to the first-order algorithm VPG. This phenomenon instead reflects that our SHSODM enjoy more stable performance. To further illustrate the robustness of SHSODM, following Zhao et al. [2022], we provide the maximal average return and the standard deviation over five trials in Table 2. A higher maximal average return indicates the algorithm has the ability to obtain the better agent. Table 2 shows SHSODM has the highest maximal average return, and outperforms other algorithms over all tested environments.

**Cost in Computing Directions.** We now compare the computational efficiency of SHSODM with SCRN, since they are both second-order algorithms. We profile the total time required to compute the update direction for the two algorithms over some representative environments. To demon-

strate, we let each algorithm run for $10^3$ epochs, and the batch size of each epoch be $10^3$ as well. Figure 2 shows that our SHSODM needs much less time than SCRN, thus it is more efficient. We remark that the result also has some theoretical guarantees. By analyzing the Hessian of the objective function, we observe that its condition number $\kappa$ is huge in each iteration, typically $10^7$. When one uses the conjugate gradient method or the gradient descent method to solve the cubic-regularization subproblem required by SCRN, the time complexity depends heavily on the Hessian's condition number [Golub and Van Loan, 2013]. This fact brings the unacceptable computational burden given the huge-condition-number nature of the Hessian. Fortunately, for SHSODM, the Lanczos-type algorithm we use to solve the eigenvalue problem at each iteration is proved to be free of condition number [Saad, 2011]. Therefore, SHSODM is more suitable to deal with such ill-conditioned problems emerging in RL.

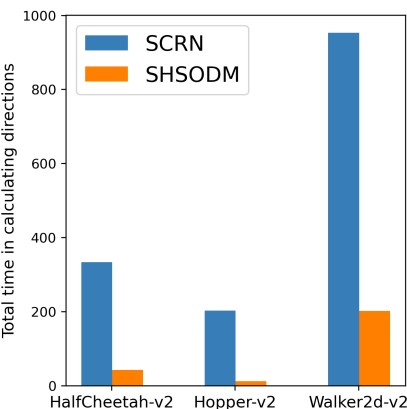

Figure 2: The $x$-axis represents three different tested environments, including `HalfCheetah-v2`, `Hopper-v2`, and `Walker2d-v2`. The $y$-axis presents the total time required to obtain the update direction in $10^3$ epochs.

## 6 CONCLUSION

Gradient dominance property enjoys wide applications in real life. In this paper, we extend HSODM from non-convex optimization to gradient-dominated stochastic world, which requires two extra customized strategies and differs from HSODM in nature. Consequently, we propose a novel stochastic second-order algorithm, SHSODM. It inherits the advantage of HSODM, only requiring solving an eigen-

value problem at each iteration, which is computationally cheaper than the cubic-regularized subproblem required by other second-order methods such as SCRN. Theoretically, we prove that SHSODM has the sample complexity matching the best-known result. Meanwhile, we demonstrate by several reinforcement learning tasks that SHSODM not only has a better and more stable performance than SCRN and other widely used algorithms in deep RL, but also is more efficient and robust in handling some ill-conditioned optimization problems in practice. For future research, one interesting direction is to apply the homogenization method to stochastic nonsmooth optimization.

## ACKNOWLEDGEMENT

The authors are grateful to the Area Chairs and the anonymous reviewers for their constructive comments. This research is partially supported by the Major Program of National Natural Science Foundation of China (Grant 72394360, 72394364).

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

# A Homogenization Approach for Gradient-Dominated Stochastic Optimization (Supplementary Material)

**Jiyuan Tan**[1,*]  **Chenyu Xue**[2,*]  **Chuwen Zhang**[2]  **Qi Deng**[3]  **Dongdong Ge**[3]  **Yinyu Ye**[1]

[*]Equal Contribution
[1]Department of Management Science and Engineering, Stanford University, Palo Alto, California, USA
[2]Research Institute for Interdisciplinary Sciences, Shanghai University of Finance and Economics, Shanghai, China
[3]Antai College of Economics and Management, Shanghai Jiao Tong University, Shanghai, China

## A  A MORE COMPREHENSIVE LITERATURE REVIEW

**First-order methods.** For the deterministic setting, Polyak et al. [1963] and Karimi et al. [2016] prove that under the PŁ condition ($\alpha = 2$), the gradient descent algorithm finds an $\epsilon$-optimal solution in $\mathcal{O}(\log(1/\epsilon))$ gradient evaluations. Then Yue et al. [2022] show that the convergence rate is optimal for the first-order algorithm. While for the stochastic setting, Khaled and Richtárik [2022] show that stochastic gradient descent (SGD) with time-varying stepsize converges to an $\epsilon$-optimal point with a sample complexity of $\mathcal{O}(1/\epsilon)$ under the PŁ condition. It is verified in Nguyen et al. [2019] that the dependency of the sample complexity of SGD on $\epsilon$ is optimal. For general gradient-dominated functions with $\alpha \in [1, 2]$, Fontaine et al. [2021] obtain a sample complexity of $O(\epsilon^{-4/\alpha+1})$ for SGD. Moreover, under the weak gradient dominance property with $\alpha = 1$, which is typically observed in reinforcement learning, it has been shown that stochastic policy gradient converges to the global optimal solution with a sample complexity of $\tilde{\mathcal{O}}(\epsilon^{-3})$ [Yuan et al., 2022].

## B  PROOF SKETCH

In this section, we give a sketch of the proof Theorem 4.1. The proof of the variance reduction version Theorem 4.2 is similar.

We first analyze Algorithm 1 and show that it can output a solution such that $\|d_k\| = \mathcal{O}(\lambda_l + \epsilon)$ in $\mathcal{O}(\log((1 + \|g_k\|)/\epsilon_{\text{ls}}\epsilon_{\text{eig}}))$, where $d_k = x_k - x_{k+1}$ (Theorem 3.1).

Then, we prove Theorem 4.1. The key of the proof is to estimate the one-step progress of SHSODM (Lemma D.2). Lemma D.2 analyzes the progress of objective value

$$F(x_k) - F(x_{k+1}) \geq \mathcal{O}(\|d_k\|^3 - \epsilon), \tag{10}$$

where $\epsilon$ is some small error cause by perturbation and randomness, and upper bounds of gradient norm $\|g_{k+1}\|$,

$$\|g_{k+1}\| \leq \mathcal{O}(\|d_k\|^2 + \lambda_k\|d_k\| + \epsilon). \tag{11}$$

Note that $\epsilon$ may not be the same in (10) and (11), but we omit the constant and use $\epsilon$ to represent small error. (11) means that if the step size $\|d_k\|$ is small, the current point is close to the optimal point. The line search (Algorithm 1) guarantees $\lambda_k = \mathcal{O}(\|d_k\| + \epsilon)$ (Theorem 3.1). Combine three equations, we get an estimation of gradient norm

$$\|g_{k+1}\| \leq \mathcal{O}(\|d_k\|^2 + \epsilon_3) \leq \mathcal{O}((F(x_k) - F(x_{k+1}))^{2/3} + \epsilon_1).$$

Now, we use the gradient dominance property.

$$F(x_{k+1}) - F^* \leq \mathcal{O}(\|g_{k+1}\|^\alpha) \leq \mathcal{O}((F(x_k) - F(x_{k+1}))^{2\alpha/3} + \epsilon) \tag{12}$$

(12) establishes a recursive relationship of the objective value. Let $\Delta_k = F(x_k) - F^*$, $\Delta_k$ satisfies

$$\Delta_{k+1} \leq C_\Delta(\Delta_k - \Delta_{k+1})^{2\alpha/3}, \tag{13}$$

where $C_\Delta$ is a constant. Lemma D.1 analyzes the convergence rate of $\Delta_k$ that satisfies (13) for different $\alpha$. Combining these results, we are able to obtain the desired conclusion.

# C TECHNICAL RESULTS FOR THEOREM 3.1 ON THE ADAPTIVE SEARCH OF $\delta_k$

Before we delve into a discussion of Theorem 3.1, we introduce some auxiliary lemmas on the two subroutines Algorithm 2 and Algorithm 1. **Since it is understood at iteration $k$, we omit subscript $k$ for conciseness**. Recall that

$$A(\delta) := \begin{bmatrix} H & g \\ g^T & -\delta \end{bmatrix}$$

and

$$\lambda(\delta) = \lambda_{\min}(A(\delta)).$$

**Lemma C.1.** *Suppose that $A(\delta)$ and $\lambda(\delta)$ are defined as above, then the following statements hold,*

1. *$\lambda(\delta)$ is non-decreasing in $\delta$ and $1$-Lipschitz.*
2. *Let $\lambda_{\min}(H)$ be the smallest eigenvalue of $H$.*

$$\lim_{\delta \to +\infty} \lambda(\delta) = +\infty, \quad \lim_{\delta \to -\infty} \lambda(\delta) = -\lambda_{\min}(H).$$

3. *Let $E_\lambda$ be the eigenspace of $H$ corresponding to the eigenvalue $\lambda$. If $g$ is not orthogonal to $E_\lambda$, that is,*

$$\mathcal{P}_{\mathcal{S}_{\min}}(g) \neq 0,$$

   *then for any $C_e > 0$, there exist $\delta_{C_e}$ such that*

$$\lambda(\delta_{C_e}) = C_e \left\| (H + \lambda(\delta_{C_e})I)^{-1} g \right\|. \tag{14}$$

*Proof.* (1) $\lambda(\delta)$ is a non-decreasing function of $\delta$ since

$$F(\delta_1) - F(\delta_2) \succeq 0, \quad \delta_1 > \delta_2$$

Let $\delta_1 > \delta_2$, $v$ be the smallest eigenvector of $F(\delta_2)$.

$$(\lambda(\delta_1) - \lambda(\delta_2))\|v\|^2 \leq v^\mathsf{T}(F(\delta_1) - F(\delta_2))v \leq (\delta_1 - \delta_2)\|v\|^2.$$

(2) $\lim_{\delta \to +\infty} \lambda(\delta) = +\infty$ since $\lambda(\delta) \geq \delta$. By Schur complement,

$$H + \lambda(\delta)I - \frac{1}{\lambda(\delta) - \delta} gg^\mathsf{T} \succeq 0.$$

For any $\epsilon > 0$, and sufficiently small $\delta$,

$$H + (-\lambda_{\min}(H) + \epsilon)I - \frac{1}{-\lambda_{\min}(H) + \epsilon - \delta} gg^\mathsf{T} \succ 0.$$

Thus, $\lambda_{\min}(H) \leq \lim_{\delta \to -\infty} \lambda(\delta) \leq \lambda_{\min}(H) + \epsilon$, for any $\epsilon > 0$, which implies $\lim_{\delta \to -\infty} \lambda(\delta) = \lambda_{\min}(H)$. Furthermore, we can get

$$\lambda(\delta) \leq -\lambda_{\min}(H) + \epsilon, \forall \delta \leq -\frac{\|g\|}{\epsilon} - \lambda_{\min}(H). \tag{15}$$

(3) Let $h(\lambda) = \lambda - C_e \left\| (H + \lambda I)^{-1} g \right\|$. Since $g$ is not orthogonal to $E_{\lambda_{\min}}$, $\lim_{\lambda \to -\lambda_{\min}(H)} h(\lambda) = -\infty$ and $\lim_{\lambda \to +\infty} h(\lambda) = +\infty$. Obviously, by the monotonicity of $\lambda(\delta)$ and 2, there exist $\delta \in \mathbb{R}$ satisfies the equation. $\square$

The following result gives a uniform upper bound and lower bound of the solution of (14)

**Lemma C.2.** *Suppose that $\|\mathcal{P}_{\mathcal{S}_{\min}}(g)\| \geq \epsilon_{eig}$, $\|H\|_2 \leq C_H$. Then $\delta_{C_e}$ and $\lambda_{C_e} = \lambda(\delta_{C_e})$ in Lemma C.1 satisfy*

$$\lambda_{C_e} \in \left[ \frac{-\lambda_{\min}(H) + \sqrt{\lambda_{\min}^2(H) + 4\epsilon_{eig}}}{2}, C_e\|g\| + C_H + 1 \right],$$

*and*

$$\delta_{C_e} \in \left[ -\frac{\|g\| \left( C_H + \sqrt{C_H + \epsilon_{eig}} \right)}{\epsilon_{eig}} - C_H, C_e\|g\| + C_H + 1 \right].$$

*Proof.* Let $h(\lambda) = \lambda - C_e \|(H + \lambda I)^{-1} g\|$, then $h(\lambda)$ is nondecresing. If $\lambda > C_e \|g\| + C_H + 1$,

$$h(\lambda) = \lambda - C_e \|(H + \lambda I)^{-1} g\| \geqslant \lambda - C_e \|g\|/(\lambda - C_H) > 0.$$

Thus, $\lambda_{C_e} \leqslant C_e \|g\| + C_H + 1$. On the other hand, if $\lambda \leqslant \frac{-\lambda_{\min}(H) + \sqrt{\lambda_{\min}^2(H) + 4\epsilon_{\text{eig}}}}{2}$,

$$h(\lambda) \leqslant \lambda - C_e \epsilon_{\text{eig}}/(\lambda - \lambda_{\min}(H)) < 0.$$

Thus, $\lambda_{C_e} \geqslant \frac{-\lambda_{\min}(H) + \sqrt{\lambda_{\min}^2(H) + 4\epsilon_{\text{eig}}}}{2}$. Notice that

$$\delta_{C_e} \leqslant \lambda(\delta_{C_e}) \leqslant C_e \|g\| + C_H + 1.$$

Let $\epsilon = \frac{\lambda_{\min}(H) + \sqrt{\lambda_{\min}^2(H) + 4\epsilon_{\text{eig}}}}{2}$ in (15), we get

$$\delta_{C_e} \geqslant -\frac{\|g\| \left(-\lambda_{\min}(H) + \sqrt{\lambda_{\min}^2(H) + 4\epsilon_{\text{eig}}}\right)}{\epsilon_{\text{eig}}} - \lambda_{\min}(H) \geqslant -\frac{\|g\| \left(C_H + \sqrt{C_H + \epsilon_{\text{eig}}}\right)}{\epsilon_{\text{eig}}} - C_H.$$

$\square$

**Proposition C.1.** *The output of Algorithm 2 satisfies $\|\mathcal{P}_{E_{\lambda_{\min}}}(g')\| \geqslant \epsilon_{eig}$ and $\|g' - g\| \leqslant \epsilon_{eig}$.*

*Proof.* If $\mathcal{P}_{E_{\lambda_{\min}}}(g) \geqslant \epsilon_{\text{eig}}$, output $g' = g$ and the two inequalities hold trivially. If $\mathcal{P}_{E_{\lambda_{\min}}}(g) < \epsilon_{\text{eig}}$,

$$g' = g + \epsilon_{\text{eig}} \frac{\mathcal{P}_{E_{\lambda_{\min}}}(g)}{\|\mathcal{P}_{E_{\lambda_{\min}}}(g)\|}.$$

We have $\|g - g'\| = \epsilon_{\text{eig}}$ and $\|\mathcal{P}_{E_{\lambda_{\min}}}(g')\| = \|\mathcal{P}_{E_{\lambda_{\min}}}(g) \left(1 + \frac{\epsilon_{\text{eig}}}{\|\mathcal{P}_{E_{\lambda_{\min}}}(g)\|}\right)\| \geqslant \epsilon_{\text{eig}}$.

$\square$

### C.0.1 Proof of Theorem 3.1

Finally, we are ready to prove Theorem 3.1.

*Proof.* If we compute a $\hat{\delta}_{C_e}$ such that

$$|\lambda(\delta_{C_e}) - \lambda(\hat{\delta}_{C_e})| \leqslant \epsilon, \tag{16}$$

then

$$\left\|(H + \lambda(\delta_{C_e})I)^{-1} g\right\| - \left\|(H + \lambda(\hat{\delta}_{C_e})I)^{-1} g\right\| \leqslant \left\|\left((H + \lambda(\delta_{C_e})I)^{-1} - (H + \lambda(\hat{\delta}_{C_e})I)^{-1}\right) g\right\|$$

$$\leqslant \epsilon \left\|(H + \lambda(\hat{\delta}_{C_e})I)^{-1}\right\| \left\|(H + \lambda(\delta_{C_e})I)^{-1}\right\| \|g\|$$

$$\leqslant \frac{4\epsilon \|g\|}{\left(-\lambda_{\min}(H) + \sqrt{\lambda_{\min}^2(H) + 4\epsilon_{\text{eig}}}\right)^2}.$$

Taking

$$\epsilon \leqslant \epsilon_{\text{ls}} \min \left\{1/2, \left(-\lambda_{\min}(H) + \sqrt{\lambda_{\min}^2(H) + 4\epsilon_{\text{eig}}}\right)^2 / (8C\|g\|)\right\},$$

then (16) guarantees

$$|\lambda(\delta_{C_e}) - \lambda(\hat{\delta}_{C_e})| \leqslant \epsilon_{\text{ls}}/2, \ C \left\|\left\|(H + \lambda(\delta_{C_e})I)^{-1} g\right\| - \left\|(H + \lambda(\hat{\delta}_{C_e})I)^{-1} g\right\|\right\| \leqslant \epsilon_{\text{ls}}/2$$

and

$$|h(\hat{\delta}_{C_e})| \leqslant \epsilon_{\text{ls}}.$$

By Lemma C.1, $\lambda(\delta)$ is 1-Lipschitz continuous. By Assumption 4.1, $f(x, \xi)$ are Lipschitz continuous. Thus, the sample Hessian are bounded. Together with Proposition C.1, the conditions in Lemma C.2 are satisfied. Lemma C.2 tells us $\delta_{C_e}$ lies in an interval of length $O\left(1 + \|g\| \epsilon_{\text{eig}}^{-1/2}\right)$. Using bisection, we can compute a proper $\hat{\delta}_{C_e}$ such that (16) holds in $\mathcal{O}(\log((1 + \|g\|)/\epsilon_{\text{ls}} \epsilon_{\text{eig}}))$.

$\square$

# D PROOF OF SAMPLE COMPLEXITY RESULTS OF ALGORITHM 3

**Lemma D.1.** *Suppose a non-negative sequence $\{\Delta_k\}$ satisfies*

$$\Delta_{k+1} \le C_\Delta (\Delta_k - \Delta_{k+1})^{2\alpha/3},$$

*where $C_e$ is a constant. For any $\epsilon > 0$,*

1. *If $\alpha = 3/2$, $\forall k \ge \mathcal{O}(\log \epsilon)$, we have $\Delta_k \le \epsilon$.*
2. *If $\alpha \in (1, 3/2)$, $\forall k \ge O\left(\epsilon^{1-3/2\alpha}\right)$, we have $\Delta_k \le \epsilon$.*
3. *If $\alpha > 3/2$, $\forall k \ge \mathcal{O}(\log \log \epsilon)$, we have $\Delta_k \le \epsilon$.*

*Proof.* Let $\beta = 2\alpha/3$.

(1)If $\beta = 1$, obviously $\Delta_k$ converge to 0 linearly as

$$\Delta_{k+1} \le \frac{1}{1 + C_\Delta} \Delta_k.$$

If $\beta \ne 1$, let $D_k = \Delta_k / C_\Delta^{1/(1-\beta)}$, then

$$D_{k+1} \le (D_k - D_{k+1})^\beta$$

(2) If $\beta < 1$, let $h(x) = x^{1-1/\beta}$,

$$h(D_{k+1}) - h(D_k) \ge (1/\beta - 1)D_k^{-1/\beta}(D_k - D_{k+1})$$
$$\ge (1/\beta - 1)(D_{k+1}/D_k)^{1/\beta}.$$

**Case I.** $D_{k+1} < \frac{1}{2}D_k$. This case will happen at most $\log(D_0/\epsilon)$ times.

**Case II.** $D_{k+1} \ge \frac{1}{2}D_k$. $h(D_{k+1}) - h(D_k) \ge (1/\beta - 1)/2^{1/\beta}$. Thus,

$$D_n < \epsilon, \forall n \ge N_\epsilon = \log(D_0/\epsilon)/\log 2 + \frac{\epsilon^{1-1/\beta} - h(D_0)}{(1/\beta - 1)/2^{1/\beta}} = O\left(\epsilon^{1-1/\beta}\right).$$

We obtain sublinear rate.

(3) If $\beta > 1$, let $N_0 = \inf\{n : D_n < 1/2\}$. Then for all $n < N_0 - 1$,

$$1/2 \le D_{k+1} \le (D_k - D_{k+1})^\beta.$$

Thus $N_0 \le 2^{1/\beta}\lceil D_0 \rceil + 1$. For all $n > N_0$,

$$D_{n+1} \le D_n^\beta \le D_{N_0}^{\beta^{N-N_0}}.$$

Combine the two cases, we have

$$D_n < \epsilon, \forall n \ge N_\epsilon = 2^{1/\beta}\lceil D_0 \rceil + 1 + (\log\log(1/\epsilon) - \log\log 2)/\log\beta = \mathcal{O}(\log\log(1/\epsilon)).$$

we obtain superlinear rate. $\square$

**Lemma D.2.** *For Algorithm 3, we have*

$$\|d_k\|^3 \le \frac{6}{L_H}\left(F(x_k) - F(x_{k+1}) + \epsilon_{eig}^{3/2} + \epsilon_{ls}^3 + \|H_k - \hat{H}_k\|^3 + \|g_k - \hat{g}_k\|^{3/2}\right). \tag{17}$$

*and*

$$\|g_{k+1}\| \le \frac{5L_H + 14}{6}\|d_k\|^2 + \epsilon_{ls}^2 + \epsilon_{eig} + \|\hat{g}_k - \hat{g}_k'\| + \frac{1}{2}\|\hat{H}_k - H_k\|^2. \tag{18}$$

*Proof.* Suppose that $(\hat{H}_k + \lambda_k I)d_k = \hat{g}'_k$, where $\hat{H}_k = \frac{1}{N_H}\sum_{i=1}^{N_H} H_{k,i}$, $\hat{g}_k = \frac{1}{N_g}\sum_{i=1}^{N_g} g_{k,i}$ is the estimation of Hessian and gradient, and $\hat{g}'_k$ is the perturbation of $\hat{g}_k$. By Proposition C.1, $\|\hat{g}_k - \hat{g}'_k\| \le \epsilon_{\text{eig}}$. Then we have,

$$
\begin{aligned}
F(x_{k+1}) - F(x_k) &\le -\langle g_k, d_k\rangle + \frac{1}{2}d_k^T H_k d_k + \frac{L_H}{6}\|d_k\|^3 \\
&= -\langle \hat{g}'_k, d_k\rangle + \frac{1}{2}d_k^T \hat{H}_k d_k + \frac{L_H}{6}\|d_k\|^3 - \langle g_k - \hat{g}'_k, d_k\rangle + \frac{1}{2}d_k^{\mathsf{T}}(H_k - \hat{H}_k)d_k \\
&\le -\frac{\lambda_k}{2}\|d_k\|^2 + \frac{2+L_H}{6}\|d_k\|^3 + \|g_k - \hat{g}'_k\|\|d_k\| + \frac{1}{2}\|d_k\|^2\|H_k - \hat{H}_k\|.
\end{aligned}
\tag{19}
$$

By Young's inequality,

$$
\|g_k - \hat{g}_k\|\|d_k\| \le \frac{2}{3}\|g_k - \hat{g}_k\|^{3/2} + \frac{1}{3}\|d_k\|^3, \ \ \|d_k\|^2\|H_k - \hat{H}_k\| \le \frac{2}{3}\|d_k\|^3 + \frac{1}{3}\|H_k - \hat{H}_k\|^3.
$$

Thus,

$$
F(x_{k+1}) - F(x_k) \le -\frac{\lambda_k}{2}\|d_k\|^2 + \frac{6+L_H}{6}\|d_k\|^3 + \frac{2}{3}\epsilon_{\text{eig}}{}^{3/2} + \frac{1}{6}\|H_k - \hat{H}_k\|^3 + \frac{2}{3}\|g_k - \hat{g}_k\|^{3/2}
$$

Thus, let $C_e = \frac{2(L_H+4)}{3}$ in Algorithm 1, by Theorem 3.1, we have

$$
\lambda_k \ge \frac{2(L_H+4)}{3}\|d_k\| - \epsilon_{\text{ls}}.
\tag{20}
$$

By Young's inequality, $\epsilon_{\text{ls}}\|d_k\|^2 \le \frac{2}{3}\|d_k\|^3 + \frac{1}{3}\epsilon_{\text{ls}}^3$,

$$
\begin{aligned}
\frac{L_H}{6}\|d_k\|^3 - \frac{\epsilon_{\text{ls}}^3}{6} &\le \frac{L_H+2}{6}\|d_k\|^3 - \frac{\epsilon_{\text{ls}}}{2}\|d_k\|^2 \\
&\le \frac{L_H+2}{6}\|d_k\|^3 + \frac{\lambda_k}{2}\|d_k\|^2 - \frac{L_H+4}{3}\|d_k\|^3 \\
&\le F(x_k) - F(x_{k+1}) + \frac{2}{3}\epsilon_{\text{eig}}{}^{3/2} + \frac{1}{6}\|H_k - \hat{H}_k\|^3 + \frac{2}{3}\|g_k - \hat{g}_k\|^{3/2},
\end{aligned}
$$

where we use (20) in the second inequality and (19) in the last inequality. We get

$$
\|d_k\|^3 \le \frac{6}{L_H}\left(F(x_k) - F(x_{k+1}) + \epsilon_{\text{eig}}{}^{3/2} + \epsilon_{\text{ls}}^3 + \|H_k - \hat{H}_k\|^3 + \|g_k - \hat{g}_k\|^{3/2}\right).
\tag{21}
$$

For gradient,

$$
\begin{aligned}
\|g_{k+1}\| &\le \|g_{k+1} - g_k - H_k d_k\| + \|g_k + H_k d_k\| \\
&\le \|g_{k+1} - g_k - H_k d_k\| + \|\hat{g}_k + \hat{H}_k d_k\| + \|g_k - \hat{g}_k\| + \|\hat{g}_k - \hat{g}'_k\| + \|\hat{H}_k - H_k\|\|d_k\| \\
&\le \frac{L_H+1}{2}\|d_k\|^2 + \lambda_k\|d_k\| + \epsilon_{\text{eig}} + \|\hat{g}_k - \hat{g}'_k\| + \frac{1}{2}\|\hat{H}_k - H_k\|^2 \\
&\le \frac{L_H+1}{2}\|d_k\|^2 + \frac{2(L_H+4)}{3}\|d_k\|^2 + \epsilon_{\text{ls}}\|d_k\| + \epsilon_{\text{eig}} + \|\hat{g}_k - \hat{g}'_k\| + \frac{1}{2}\|\hat{H}_k - H_k\|^2 \\
&\le \frac{7L_H+19}{6}\|d_k\|^2 + \epsilon_{\text{ls}}^2 + \epsilon_{\text{eig}} + \|\hat{g}_k - \hat{g}'_k\| + \frac{1}{2}\|\hat{H}_k - H_k\|^2
\end{aligned}
$$

where we have used $\lambda_k \le \frac{L_H+4}{3}\|d_k\| + \epsilon_{\text{ls}}$ in the fourth inequality. This completes the proof. $\square$

## D.1   PROOF OF THEOREM 4.1

Now we use the previous results to complete the proof regarding the convergence rate.

**Theorem D.1.** *Suppose that $F(x)$ satisfies the gradient dominance assumption with index $\alpha$, Assumption 4.1. The output of Algorithm 3 with parameters $\epsilon_{eig}, \epsilon_{ls}, \epsilon_{noise}$ and $K$, satisfies the following statements,*

- *If $\alpha \in [1, 3/2)$, $F(x_K) - F^* \le O\left(K^{\frac{-2\alpha}{3-2\alpha}} + \epsilon_{eig}^\alpha + \epsilon_{ls}^{2\alpha} + 2\epsilon_{noise}^\alpha\right)$.*

- If $\alpha = 3/2$, $F(x_K) - F^* \leq O\left(\exp(-K) + \epsilon_{eig}^\alpha + \epsilon_{ls}^{2\alpha} + 2\epsilon_{noise}^\alpha\right)$.
- If $\alpha \in (3/2, 2]$, $F(x_K) - F^* \leq O\left(\exp(\exp(-K)) + \epsilon_{eig}^\alpha + \epsilon_{ls}^{2\alpha} + 2\epsilon_{noise}^\alpha\right)$.

*Proof.* By the gradient dominance assumption and (18),

$$
\begin{aligned}
F\left(x^{k+1}\right) - F^* &\leqslant C_{\text{gd}}\|g_{k+1}\|^\alpha \\
&\leqslant C_{\text{gd}}\left(\frac{7L_H + 19}{6}\|d_k\|^2 + \epsilon_{ls}^2 + \epsilon_{eig} + \|\hat{g}_k - \hat{g}_k'\| + \frac{1}{2}\|\hat{H}_k - H_k\|^2\right)^\alpha
\end{aligned}
$$

Note that for any $x, y > 0$, we have

$$
(x + y)^r \leqslant \begin{cases} x^r + y^r & , r \in (0, 1), \\ 2^{r-1}\left(x^r + y^r\right) & , r \geqslant 1. \end{cases}
$$

Thus, $(x + y)^r = O\left(x^r + y^r\right)$ and

$$
\begin{aligned}
F\left(x^{k+1}\right) - F^* &\leqslant O\left(\|d_k\|^{2\alpha} + \epsilon_{ls}^{2\alpha} + \epsilon_{eig}^\alpha + \|\hat{g}_k - \hat{g}_k'\|^\alpha + \frac{1}{2}\|\hat{H}_k - H_k\|^{2\alpha}\right) \\
&\leqslant O\left(F\left(x^k\right) - F\left(x^{k+1}\right) + \epsilon_{eig}^{3/2} + \epsilon_{ls}^3 + \|H_k - \hat{H}_k\|^3 + \|g_k - \hat{g}_k\|^{3/2}\right)^{2\alpha/3} \\
&\quad + O\left(\epsilon_{ls}^{2\alpha} + \epsilon_{eig}^\alpha + \|\hat{g}_k - \hat{g}_k'\|^\alpha + \frac{1}{2}\|\hat{H}_k - H_k\|^{2\alpha}\right) \\
&= O\left(\left(F\left(x^k\right) - F\left(x^{k+1}\right)\right)^{2\alpha/3} + \epsilon_{eig}^\alpha + \epsilon_{ls}^{2\alpha} + \|H_k - \hat{H}_k\|^{2\alpha} + \|g_k - \hat{g}_k\|^\alpha\right),
\end{aligned}
$$

where we have used (17) in the inequality. Take expectation, we get

$$
\mathbb{E}\left[F\left(x^{k+1}\right) - F^*\right] \leqslant O\left(\left(\mathbb{E}\left[F\left(x^k\right) - F\left(x^{k+1}\right)\right]\right)^{2\alpha/3} + \epsilon_{eig}^\alpha + \epsilon_{ls}^{2\alpha} + \left(\mathbb{E}\left[\|g_k - \hat{g}_k\|^2\right]\right)^{\alpha/2} + \mathbb{E}\left[\|\hat{H}_k - H_k\|^{2\alpha}\right]\right)
$$

$$(22)$$

Suppose we take $N_H = O\left(\epsilon_{noise}^{-1}\right)$, $N_g = O\left(\epsilon_{noise}^{-2}\right)$, by Lemma 3 in Masiha et al. [2022] and Assumption 4.2,

$$
\mathbb{E}\left[\|g_k - \hat{g}_k\|^2\right] \leq \epsilon_{noise}^2, \quad \mathbb{E}\left[\|\hat{H}_k - H_k\|^{2\alpha}\right] \leq \epsilon_{noise}^\alpha,
$$

Let $\Delta_k = \mathbb{E}[F\left(x_k\right)] - F^* - \left(\epsilon_{eig}^\alpha + \epsilon_{ls}^{2\alpha} + 2\epsilon_{noise}^\alpha\right)$, there exist $C_\Delta$ such that

$$
\Delta_{k+1} \leq C_\Delta(\Delta_k - \Delta_{k+1})^{2\alpha/3}.
$$

Applying Lemma D.1, we obtain the result. $\qquad\square$

*Proof of Theorem 4.1.* Finally, we prove the sample complexity result in Theorem 4.1. For a given tolerance $\epsilon$, let $\epsilon_{eig} = O\left(\epsilon^{1/\alpha}\right)$, $\epsilon_{ls} = O\left(\epsilon^{1/2\alpha}\right)$, $\epsilon_{noise} = O\left(\epsilon^{1/\alpha}\right)$ in Theorem D.1. We need $O\left(\epsilon^{2/\alpha}\right)$ samples in each iteration. Thus, by Lemma D.1, we have the following sample complexity results.

For time complexity, note that in each iteration, by Theorem 3.1, we need $\mathcal{O}(\log((1 + \|g\|)/\epsilon_{ls}\epsilon_{eig}))$ step in Algorithm 1. From (18) and Theorem D.1, we know $\mathbb{E}\|g_k\| \leqslant \mathcal{O}(\epsilon^{1/\alpha} + K^{-\frac{4\alpha}{3-2\alpha}})$. Therefore, $\mathbb{E}[\mathcal{O}(\log((1 + \|g\|)/\epsilon_{ls}\epsilon_{eig}))] \leq \mathcal{O}(\log(1/\epsilon))$.

| $\alpha$ | Expected Time Complexity | Sample Complexity |
|---|---|---|
| $\alpha \in [1, 3/2)$ | $O\left(\log(1/\epsilon)\epsilon^{-7/(2\alpha)+3/4}\right)$ | $O\left(\epsilon^{-7/(2\alpha)+1}\right)$ |
| $\alpha = 3/2$ | $O\left(\log^2(1/\epsilon)\epsilon^{-2/\alpha-1/4}\right)$ | $O\left(\log(1/\epsilon)\epsilon^{-2/\alpha}\right)$ |
| $\alpha \in (3/2, 2]$ | $O\left(\log(1/\epsilon)\log\log(1/\epsilon)\epsilon^{-2/\alpha-1/4}\right)$ | $O\left(\log\log(1/\epsilon)\epsilon^{-2/\alpha}\right)$ |

$\qquad\square$

## D.2 PROOF OF COROLLARY 4.1

If $F$ only satisfies the weak gradient dominance property (2.1), notice that Lemma D.2 still holds. Following the proof of Theorem D.1, we can get a variant of (22),

$$\mathbb{E}\left[F\left(x^{k+1}\right) - F^*\right] \leqslant O\left(\left(\mathbb{E}\left[F\left(x^k\right) - F\left(x^{k+1}\right)\right]\right)^{2\alpha/3} + \epsilon_{\text{eig}}^\alpha + \epsilon_{\text{ls}}^{2\alpha} + \left(\mathbb{E}\left[\|g_k - \hat{g}_k\|^2\right]\right)^{\alpha/2} + \mathbb{E}\left[\|\hat{H}_k - H_k\|^{2\alpha}\right] + \epsilon_{\text{noise}}\right)$$

Let $\Delta_k' = F\left(x_k\right) - F^* - \left(\epsilon_{\text{eig}}^\alpha + \epsilon_{\text{ls}}^{2\alpha} + 2\epsilon_{\text{noise}}^\alpha + \epsilon_{\text{weak}}\right)$, there exist $C_\Delta'$ such that

$$\Delta_{k+1}' \leq C_\Delta'(\Delta_k' - \Delta_{k+1}')^{2\alpha/3}.$$

Applying Lemma D.1, we get the result.

## E PROOF OF SAMPLE COMPLEXITY RESULTS OF ALGORITHM 4

In this section, we strengthen the sample complexity result using variance reduction techniques. (22) implies that the the estimation error $\mathbb{E}[\|g_k - \hat{g}_k\|^2]$ and $\mathbb{E}[\|H_k - \hat{H}_k\|^{2\alpha}]$ directly influence the convergence rate. We need to use the sample more cleverly to increase sample complexity. By Assumption 4.1, we have $\|H_k - \hat{H}_k\| \leqslant C_H = 2L_g$, we have,

$$\mathbb{E}\left[\|H_k - \hat{H}_k\|^{2\alpha}\right] \leqslant C_H^{2\alpha}\mathbb{E}\left[\|H_k - \hat{H}_k\|^2/C_H^2\right] \leqslant C_H^\alpha \mathbb{E}\left[\|H_k - \hat{H}_k\|^2\right]^{\alpha/2}.$$

Thus, the noise error caused by Hessian estimation is of the same order as gradient estimation and we only need to analyze the impact of applying variance reduction techniques to the gradient. A similar analysis applies to Hessian too.

We mainly use the variance reduction technique from Fang et al. [2018]. Recall that we use gradient estimation in Algorithm 4,

$$v_k = \begin{cases} \nabla_{S_k} F(x_k), & k \bmod K_C = 0, \\ \nabla_{S_k} F(x_k) - \nabla_{S_k} F(x_{k-1}) + v_{k-1}, & k \bmod K_C \neq 0, \end{cases}$$

where $\nabla_{S_k} F(x_k) = \frac{1}{|S_k|}\sum_{k=1}^{|S_k|} \nabla F(x_k, \xi_k)$. Let

$$e_k = \begin{cases} \nabla_{S_k} F(x_k) - F(x_k), & k \bmod K_C = 0, \\ \nabla_{S_k} F(x_k) - \nabla_{S_k} F(x_{k-1}) - (F(x_k) - F(x_{k-1})), & k \bmod K_C \neq 0, \end{cases}$$

we have error decomposition $F(x_k) - v_k = \sum_{i=\lfloor k/K_C \rfloor K_C}^{k} e_i$. We define the sigma field $\mathcal{F}_k = \sigma(x_0, \cdots, x_k, e_1, \cdots, e_{k-1})$. The following Lemma analyzes the error $e_k$.

**Lemma E.1.**
$$\mathbb{E}\left[\|e_k\|^2|\mathcal{F}_k\right] \leqslant \frac{4L_g}{n_{k,g}}\|d_k\|^2, \qquad k \bmod K_C \neq 0$$

*where $d_k = x_k - x_{k-1}$.*

*Proof.* We have

$$\mathbb{E}\left[\|e_k\|^2|\mathcal{F}_k\right] = \mathbb{E}\left[\left\|\frac{1}{n_{k,g}}\sum_{i=1}^{n_{k,g}} \nabla F(x_k, \xi_i) - \nabla F(x_{k-1}, \xi_i) - (F(x_k) - F(x_{k-1}))\right\|^2\right]$$

$$= \frac{1}{n_{k,g}}\mathbb{E}\left[\|\nabla F(x_k, \xi_1) - \nabla F(x_{k-1}, \xi_1) - (F(x_k) - F(x_{k-1}))\|^2\right]$$

$$\leqslant \frac{2}{n_{k,g}}\mathbb{E}\left[\|\nabla F(x_k, \xi_1) - \nabla F(x_{k-1}, \xi_1)\|^2\right] + \frac{2}{n_{k,g}}\mathbb{E}\left[\|F(x_k) - F(x_{k-1})\|^2\right]$$

$$\leqslant \frac{4L_g}{n_{k,g}}\|x_k - x_{k-1}\|^2 = \frac{4L_g}{n_{k,g}}\|d_k\|^2.$$

$\square$

*Proof of Theorem 4.2.* Using Lemma E.1, we can estimate the variance of the gradient estimation,

$$\mathbb{E}\left[\|\nabla F(x_k) - v_k\|^2\right] = \mathbb{E}\left[\left\|\sum_{i=\lfloor k/K_C \rfloor K_C}^{k} e_i\right\|^2\right] = \sum_{i=\lfloor k/K_C \rfloor K_C}^{k} \mathbb{E}\left[\|e_i\|^2\right]$$

$$\leqslant \frac{\sigma^2}{n_{\lfloor k/K_C \rfloor K_C, g}} + 4L_g \sum_{i=\lfloor k/K_C \rfloor K_C + 1}^{k} \mathbb{E}\left[\frac{\|d_i\|^2}{n_{k,g}}\right]$$

For $\alpha \in [1, 3/2)$, if we take

$$n_{k,g} = \begin{cases} O\left(k^{4/(3-2\alpha)}\right) & k \mod K_C = 0, \\ O\left(\|d_k\|^2 K_C(\lfloor k/K_C \rfloor K_C)^{4/(3-2\alpha)}\right) & k \mod K_C \neq 0, \end{cases}$$

we have $\mathbb{E}\left[\|F(x_k) - v_k\|^2\right]^{\alpha/2} \leqslant O\left(1/k^{1/(3/(2\alpha)-1)}\right)$. Follow the proof of Theorem D.1, we can get $F\left(x^{kK_C}\right) - F^* = O\left((kK_C)^{-2\alpha/(3-2\alpha)}\right)$.

Next, we estimate the expected sample complexity. By Lemma D.2 we have,

$$\sum_{i=(k-1)K_C}^{kK_C-1} \|d_i\|^2 \leqslant \sum_{i=(k-1)K_C}^{kK_C-1} \left(\frac{6}{L_H} F\left(x^i\right) - F\left(x^{i+1}\right) + \epsilon_{\text{eig}}^{3/2} + \epsilon_{\text{ls}}^3 + \|H_i - \hat{H}_i\|^3 + \|g_i - \hat{g}_i\|^{3/2}\right)^{2/3}$$

$$\leqslant O\left(\mathbb{E}\left[\sum_{i=(k-1)K_C}^{kK_C-1} \left(F\left(x^i\right) - F\left(x^{i+1}\right)\right)^{2/3} + \epsilon_{\text{eig}} + \epsilon_{\text{ls}}^2 + \|H_i - \hat{H}_i\|^2 + \|g_i - \hat{g}_i\|\right]\right)$$

$$\leqslant O\left(K_C^{1/3} \left(F\left(x^{(k-1)K_C}\right) - F\left(x^{kK_C}\right)\right)^{2/3} + K_C(kK_C)^{-2/(3-2\alpha)}\right)$$

$$= O\left(K_C^{1/3} \cdot (kK_C)^{-4\alpha/(9-6\alpha)} + K_C(kK_C)^{-2/(3-2\alpha)}\right),$$

where we use $F\left(x^{kK_C}\right) - F^* = O\left((kK_C)^{-2\alpha/(3-2\alpha)}\right)$ by Theorem D.1. We get

$$E\left[\sum_{i=(k-1)K_C}^{kK_C-1} \|d_i\|^2\right] = O\left(K_C^{1/3}(K_Ck)^{\frac{-4\alpha}{3(3-2\alpha)}}\right).$$

Thus,

$$\mathbb{E}\left[\sum_{k=1}^{K} n_{k,g}\right] = O\left(\sum_{k=1}^{\lceil K/K_C \rceil} (K_Ck)^{\frac{4}{3-2\alpha}} + K_C^{4/3} \sum_{k=1}^{\lceil K/K_C \rceil} \frac{(kK_C)^{4/(3-2\alpha)}}{(K_Ck)^{4\alpha/(3(3-2\alpha))}}\right)$$

$$= O\left(K^{1+4/(3-2\alpha)}/K_C + K_C^{1/3}K^{1+(12-4\alpha)/(3(3-2\alpha))}\right).$$

Take $K_C = \mathcal{O}(K)$, use $\mathcal{O}(K) = O\left(\epsilon^{1-3/(2\alpha)}\right)$, we get the result. We need $O\left(\epsilon^{1-3/(2\alpha)}\right)$ iteration to reach $\epsilon$-approximate point. The expected sample complexity is $O\left(K^{4/(3-2\alpha)}\right) = O\left(\epsilon^{-2/\alpha}\right)$. $\qquad \square$

# F A BRIEF INTRODUCTION OF REINFORCEMENT LEARNING

A MDP $\mathcal{M}$ is specified by tuple $(\mathcal{S}, \mathcal{A}, \mathbb{P}, r, \gamma, \rho)$, where $\mathcal{S}$ is the state space; $\mathcal{A}$ is the action space; $P : \mathcal{S} \times \mathcal{A} \mapsto \Delta(\mathcal{S})$ is the transition function with $\Delta(\mathcal{S})$ the space of probability distribution over $\mathcal{S}$, and $P(s' \mid s, a)$ denotes the probability of transitioning into state $s'$ upon taking action $a$ in state $s$; $r : \mathcal{S} \times \mathcal{A} \mapsto [0, 1]$ is the reward function, and $r(s, a)$ is the immediate reward associated with taking action $a$ in state $s$; $\gamma \in (0, 1)$ is the discount factor; $\rho \in \Delta(\mathcal{S})$ is the initial state distribution. At $k$-th step, the agent is at state $s_k$ and pick one action from the action space $a_k \in \mathcal{A}$. Then, the environment gives reward $r_k$ to the agent and transit to the next state $s_{k+1}$ with probability $P(s_{k+1}|s_k, a_k)$.

The parametric policy $\pi_\theta$ is a probability distribution over $\mathcal{S} \times \mathcal{A}$ with parameter $\theta \in \mathbb{R}^d$, and $\pi_\theta(a \mid s)$ denotes the probability of taking action $a$ at a given state $s$. For example, one may consider the sofemax policy, given by

$$\pi_\theta(a \mid s) = \frac{\exp(\theta_{s,a})}{\sum_{a' \in \mathcal{A}} \exp(\theta_{s,a'})}$$

where the parameter space is $\theta \in \mathbb{R}^{|\mathcal{S}||\mathcal{A}|}$. Let $\tau = \{s_t, a_t\}_{t \geq 0}$ be the trajectory generated by the policy $\pi_\theta$, and $p(\tau \mid \pi_\theta)$ be the probability of the trajectory $\tau$ being sampled from $\pi_\theta$. Then, we have

$$p(\tau \mid \pi_\theta) = \rho(s_0) \prod_{t=0}^{\infty} \pi_\theta(a_t \mid s_t) P(s_{t+1} \mid s_t, a_t),$$

and the expected return of $\pi_\theta$ is

$$J(\pi_\theta) := \mathbb{E}_{\tau \sim p(\cdot \mid \pi_\theta)} \left[ \sum_{t=0}^{\infty} \gamma^t r(s_t, a_t) \right].$$

Assume that $\pi_\theta$ is differentiable with respect to $\theta$, and denote $J(\theta) = J(\pi_\theta)$ for simplicity. The goal of reinforcement learning is to find

$$\theta^* = \arg\max_\theta J(\theta).$$

However, $J(\theta)$ is differentiable but non-concave in general, leading to the difficulty to find the global optimal solution. With two common assumptions, including non-degenerate Fisher matrix [Agarwal et al., 2021, Yuan et al., 2022] and transferred compatible function approximation error [Agarwal et al., 2021], one can prove that

$$J^* - J(\theta) \leq \tau \|\nabla J(\theta)\| + \epsilon'$$

where $J^* := \max J(\theta)$. Hence, $J(\theta)$ satisfies the weak gradient dominance property with $\alpha = 1$, as well as Assumption 4.1 and Assumption 4.2. We list the two assumptions here for self-completeness, and the proof can be found in [Masiha et al., 2022, Yuan et al., 2022].

**Assumption F.1** (Fisher-non-degenerate.). *For all $\theta \in \mathbb{R}^d$, there exists $\mu_F > 0$ such that the Fisher information matrix $F_\rho(\theta)$ induced by policy $\pi_\theta$ and initial distribution $\rho$ satisfies*

$$F_\rho(\theta) := \mathbb{E}_{(s,a) \sim v_\rho^{\pi_\theta}} \left[ \nabla_\theta \log \pi_\theta(a \mid s) \nabla_\theta \log \pi_\theta(a \mid s)^T \right] \succeq \mu_F I_{d \times d},$$

*where $v_\rho^{\pi_\theta}(s, a) := (1 - \gamma) \mathbb{E}_{s_0 \sim \rho} \sum_{t=0}^{\infty} \gamma^t \mathbb{P}(s_t = s, a_t = a \mid s_0, \pi_\theta)$ is the state-action visitation measure.*

**Assumption F.2** (Transferred compatible function approximation error). *For all $\theta \in \mathbb{R}^d$, there exists $\epsilon_{bias} > 0$ such that the transferred compatible function approximation error with $(s, a) \sim v_\rho^{\pi_{\theta^*}}$ satisfies*

$$\mathbb{E}\left[ \left( A^{\pi_\theta}(s, a) - (1 - \gamma) u^{*T} \nabla_\theta \log \pi_\theta(a \mid s) \right)^2 \right] \leq \epsilon_{bias},$$

*where $v_\rho^{\pi_{\theta^*}}$ is the state-action distribution induced by an optimal policy, and $u^* = (F_\rho(\theta))^\dagger \nabla J(\theta)$.*

We remark that Masiha et al. [2022] also uses this reinforcement learning setting to study the performance of SCRN for the gradient-dominated function with $\alpha = 1$, which shows that our numerical experiment's setting is standard. Practically, we cannot compute $J(\theta)$ due to the infinite horizon length. Hence, we resort to truncated trajectories with the horizon length $H$, and focus on

$$\max_\theta \ J_H(\theta) := \mathbb{E}_{\tau \sim p(\cdot \mid \pi_\theta)} \left[ \sum_{t=0}^{H-1} \gamma^t r(s_t, a_t) \right].$$

We seek for the stationary point $\hat{\theta}$, that is,

$$\|\nabla J_H(\hat{\theta})\| \leq \epsilon.$$

Note that

$$\nabla J_H(\theta) = \mathbb{E}_{\tau \sim p(\cdot \mid \pi_\theta)} \left[ \sum_{h=0}^{H-1} \Psi_h(\tau) \nabla \log \pi_\theta(a_h \mid s_h) \right]$$

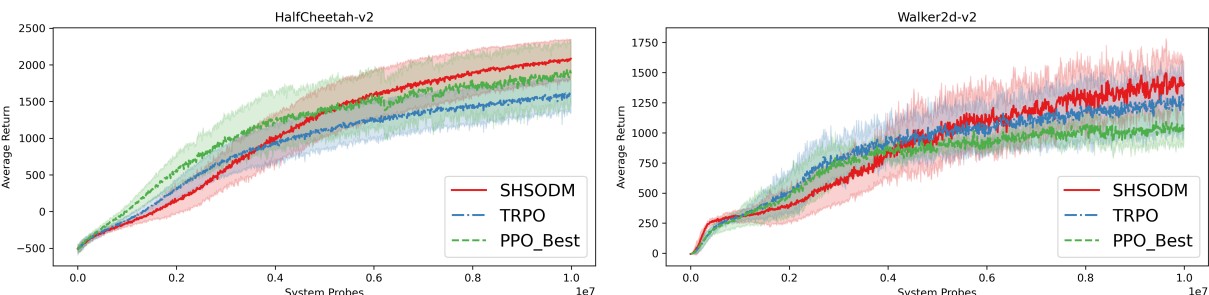

Figure 3: The $x$-axis and $y$-axis represent respectively system probes and the average return. The solid curves depict the mean values of five independent simulations, while the shaded areas correspond to the standard deviation.

where $\Psi_h(\tau) = \sum_{t=h}^{H-1} \gamma^t r(s_t, a_t)$. In practice, we can not compute the full gradient because of the failure to average over all possible trajectories $\tau$. Thus, we construct an empirical estimate by sampling different trajectories. Suppose that we sample $m$ trajectories $\tau^i = \left\{s_t^i, a_t^i\right\}_{0 \le t \le H}, 1 \le i \le m$. Then the resulting unbiased gradient estimator is

$$\hat{\nabla} J_H(\theta) = \frac{1}{m} \sum_{i=1}^{m} \sum_{h=0}^{H-1} \Psi_h\left(\tau^i\right) \nabla \log \pi_\theta\left(a_h^i \mid s_h^i\right),$$

The vanilla policy gradient (VPG) updates $\theta$ by $\theta \leftarrow \theta + \eta \hat{\nabla} J_H(\theta)$, where $\eta > 0$ is the step size.

When considering the second-order information, we have

$$\nabla^2 J_H(\theta) = \mathbb{E}_{\tau \sim p(\cdot|\pi_\theta)} \left[\nabla \Phi(\theta; \tau) \nabla \log p\left(\tau \mid \pi_\theta\right)^T + \nabla^2 \Phi(\theta; \tau)\right]$$

where $\Phi(\theta; \tau) = \sum_{h=0}^{H-1} \sum_{t=h}^{H-1} \gamma^t r(s_t, a_t) \log \pi_\theta(a_h \mid s_h)$. As a result, for trajectories $\tau^i = \left\{s_t^i, a_t^i\right\}_{t \ge 0}, 1 \le i \le m$, we have the following unbiased estimator of Hessian matrix $\nabla^2 J_H(\theta)$,

$$\hat{\nabla}^2 J_H(\theta) = \frac{1}{m} \sum_{i=1}^{m} \nabla \Phi\left(\theta; \tau^i\right) \nabla \log p\left(\tau^i \mid \pi_\theta\right)^T + \nabla^2 \Phi\left(\theta; \tau^i\right)$$

With $\hat{\nabla} J_H(\theta)$ and $\hat{\nabla}^2 J_H(\theta)$ in hand, we can use our stochastic HSODM to find $\hat{\theta}$.

## G  ADDITIONAL EXPERIMENTS

In this section, we further compare the performance of SHSODM with PPO in several RL tasks. For clean demonstration, we only include TRPO as another benchmark. When running PPO, we let its key parameter `lc_clip_range` be $\{0.1, 0.2, 0.4, 0.6, 0.8\}$, and plot the one with the best performance on average return. It is clear that our SHSODM even has better final performance than PPO, although it grows slower than PPO at the initial stage. One of the possible explanations is our SHSODM uses the second-order information of the objective function. While PPO and TRPO both use the second-order information of the constraint (TRPO involves the constraint that the consecutive two policies should not be far away from each other, and PPO penalizes this constraint in the objective function). The information of objective function maybe more useful and can induce better policies.

When comparing the time spent on calculating the update direction, we include TRPO as a benchmark, the variant of PPO. The reason is that PPO uses the first-order optimizer, while TRPO can be seen as a "second-order" method. It can be seen that our SHSODM updates faster than TRPO over the 2 tested environments and is consistently better than SCRN.

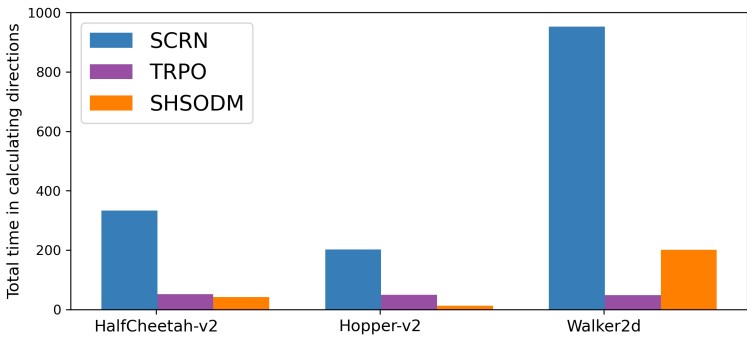

Figure 4: The $x$-axis represents three different tested environments, including `HalfCheetah-v2`, `Hopper-v2`, and `Walker2d-v2`. The $y$-axis presents the total time required to obtain the update direction in $10^3$ epochs.