# OpenReview forum: "A Homogenization Approach for Gradient-Dominated Stochastic Optimization"
_auai.org/UAI/2024/Conference — UAI 2024 poster_

### Official Review · Reviewer_wt98 · 2024-03-09

**Q2-1 Originality-Novelty:** 4
**Q2-2 Correctness-Technical Quality:** 4
**Q2-5 Clarity Of Writing:** 3

**Q1 Summary And Contributions:**

The paper introduces the Stochastic Homogeneous Second-Order Descent Method (SHSODM), a new optimization technique for stochastic functions that exhibit the gradient dominance property, a key to ensuring global convergence in non-convex optimization. This method, grounded in a novel homogenization approach, offers theoretical improvements including a detailed sample complexity analysis and enhancements through variance reduction. SHSODM stands out by matching the top sample complexity results of similar second-order methods without the need for cubic regularization, achieving lower computational costs and enhanced robustness in ill-conditioned scenarios. Empirical tests, especially in reinforcement learning tasks, confirm SHSODM's superior performance over existing alternatives.

**Q2-3 Extent To Which Claims Are Supported By Evidence:**

4: Excellent: all claims are supported by very convincing evidence (in the form of comprehensive experimental evaluation, rigorous mathematical proofs, detailed (pseudo-)code, precise references, well-motivated and realistic assumptions) and the authors deliver what they promise.

**Q2-4 Reproducibility:**

2: Fair: key resources (e.g. proofs, code, data) are unavailable but key details (e.g. proof sketches, experimental setup) are sufficiently well-described for an expert to confidently reproduce the main results.

**Q3 Main Strengths:**

1. The paper has a solid theoretical analysis.

2. The motivation is clear.

**Q4 Main Weakness:**

1. The authors provided a time complexity analysis in the paper. However, in experiments, it is not clear what the real-time complexity needed to run the algorithms.

**Q5 Detailed Comments To The Authors:**

The experiments can be refined.

**Q9 Complying With Reviewing Instructions:**

Yes

---

> ### Author Rebuttal · Authors · 2024-04-05
>
> We sincerely appreciate your feedback and are grateful for the thorough reading and valuable suggestions you provided for our paper. The extensive review comments are crucial for enhancing our future revision. We hope our responses below will adequately address your concerns and questions.
>
> **1 It is not clear what the real-time complexity needed to run the algorithms.**
>
> Thanks for your suggestion. We do not compare the runtimes of these algorithms in the paper, since we mainly concern the sample complexity. In other words, we aim to train an agent as good as possible with the same amount of system probes. Therefore, we plot the number of system probes v.s. the average return in Figure 1. The results have already shown that our algorithm can train a better agent than other algorithms.
>
> Upon your request, we now give the average returns and the average runtimes of different algorithms under the tested environments after $10^6$ system probes. The figure in the bracket represents the average return, and the outside one corresponds to the average runtime in second.
> |                | SHSODM          | SCRN            | PPO              | TRPO            | VPG             |
> |----------|-----------------|-----------------|------------------|-----------------|-----------------|
> | HalfCheetah-v2 | 2256.9 (2083.0) | 1837.6 (576.79) | 11573.0 (1881.9) | 3186.5 (1562.7) | 1211.5 (736.67) |
> | Walker2d-v2    | 2559.1 (1402.4) | 1518.6 (421.37) | 15215.0 (1043.2) | 1047.6 (1237.0) | 1949.2 (840.03) |
>
> In our experiments, different algorithms spend different time sampling state-action pairs, thus each algorithm's runtime does not reflect its true computational complexity, which is also the reason we do not compare them in the paper. We would like to compare the runtimes of different algorithms in a better way in the revision, such as using the same offline state-action pair dataset to avoid the online sampling procedure.
>
> We again thank the reviewer for the efforts in reviewing our paper and the constructive comments!

---

### Official Review · Reviewer_59MH · 2024-03-21

**Q2-1 Originality-Novelty:** 2
**Q2-2 Correctness-Technical Quality:** 3
**Q2-5 Clarity Of Writing:** 3

**Q1 Summary And Contributions:**

They proposed a new stochastic homogeneous second-order descent method (SHSODM) for the objective functions satisfying $\alpha$-gradient-dominance and proved that SHSODM matches the best-known sample complexity by other 2nd-order methods but without cubic regularization.

**Q2-3 Extent To Which Claims Are Supported By Evidence:**

3: Good: the main claims are supported by convincing evidence (in the form of adequate experimental evaluation, proofs, (pseudo-)code, references, assumptions).

**Q2-4 Reproducibility:**

4: Excellent: key resources (e.g. proofs, code, data) are available and key details (e.g. proof sketches, experimental setup) are comprehensively described for competent researchers to confidently and easily reproduce the main results.

**Q3 Main Strengths:**

They proved a convergence to the global optima for a second-order method without cubic-regularisation and it matches with the best known sample complexity. They developed a perturbation strategy to adapt HSODM to the case when function is gradient dominated.

**Q4 Main Weakness:**

Could you please address the following comments?
1) In “Nevertheless, a common drawback of these second-order methods lies in their dependence on the expensive $O(n^{3})$ computational cost at each iteration to obtain an approximate solution to an inevitable cubic-regularized subproblem.” Could you please elaborate on why $O(n^{3})$ computational cost requires? Consider the case that we are using (perturbed) gradient descent for solving sub-problem in CRN or SCRN and we do not need to compute Hessian inverse.
2) Could you please explain what the linear system is in CRN or SCRN in Table 1? “Free from solving linear system: whether the algorithm needs to solve linear systems at each iteration”
3) In Table 1, I think that $O(\epsilon^{-2})$ is the sample complexity of VR-SCRN when $\alpha=1$ in [Masiha et al. 22] but I think that it can be extended straightforwardly for $\alpha\in[1,3/2)$.
4) Why does the bound $O(\epsilon^{-7/2\alpha+3/2})$ for VR-SHSODM does not match $O(\epsilon^{-4/3}\log(1/\epsilon))$ when alpha approaches 3/2? I did not check the proof of Theorem 4.2 but it does not make sense at all. Why did not you mention this result in the contribution part of the introduction?
5) Do you need Lipschitzness of the gradient estimators in Theorem 4.1. I think that you do not use the variance reduction technique in SHSODM and the Lip. of the gradient estimators is usually used for the variance reduction technique. For example, the analysis of CRN or SCRN does not need to Lipschitzness of the gradient estimators (or even Lipschitzness of the true gradient).
6) In the experiment part, you claimed that the empirical examples you considered in your paper served as a standard grad-dom. scenario. I think that (alpha=1)-grad-dom. assumption does hold for some specific policy parameterizations.  "In this section, we evaluate the empirical performance of our SHSODM in the context of RL, which serves as a standard scenario for gradient-dominated stochastic optimization [Masiha et al., 2022]." Did you prove that the expected return in those empirical examples satisfy (alpha=1)-grad-dom.?
7) Could you please give some intuition why SHSODM outperforms TRPO and even PPO?

**Q5 Detailed Comments To The Authors:**

All is included in Q4.

**Q9 Complying With Reviewing Instructions:**

Yes

---

> ### Author Rebuttal · Authors · 2024-04-05
>
> We sincerely appreciate your feedback and are grateful for the thorough reading and valuable suggestions you provided for our paper. The extensive review comments are crucial for enhancing our future revision. We hope our responses below will adequately address your concerns and questions.
>
> **1 On $O\left(n^3\right)$ computational cost required by CRN or SCRN**
>
> We totally understand the reviewer's concern, and are sorry for the confusing words. When solving the subproblem required by CRN or SCRN, one can use the gradient descent method; see [1]. While in our context, we want to compare our algorithm with the most straightforward method to solve the linear system induced from the CRN's subproblem, i.e., computing the matrix inverse directly. We will state this more clearly in the revision. Meanwhile, our algorithm has a better dependence on the condition number than the gradient descent method, implying it is suitable to handle ill-conditioned problems such as the tested reinforcement learning tasks. We also emphasized this in the experiment section.
>
> **2 Explain what the linear system is in CRN or SCRN in Table 1? Whether the algorithm needs to solve linear systems at each iteration**
>
> Thanks for your feedback. In CRN, the algorithm needs to solve the cubic regularized problem of the following form at each iteration,
> \begin{equation*}
>     \min_{x \in \mathbb{R}^d} h(x) := \frac{1}{2} x^T A x +  b^T x + \frac{\rho}{3}||x||^3
> \end{equation*}
> By the optimality condition, it boils down to solving the linear system of the form $ (A+\lambda I)x=-b, \lambda = \rho ||x||$ for some $\lambda$; see [2].
>
>
> **3 In Table 1, $O\left(\epsilon^{-2}\right)$ is the sample complexity of VR-SCRN when $\alpha=1$ in [Masiha et al. 22] but I think that it can be extended straightforwardly for $\alpha \in[1,3 / 2)$**
>
> We agree with the reviewer, and sorry for the typo in Table 1. We will correct this in the revision.
>
> **4 Why does the bound $O\left(\epsilon^{-7 / 2 \alpha+3 / 2}\right)$ for VR-SHSODM does not match $O\left(\epsilon^{-4 / 3} \log (1 / \epsilon)\right)$ when alpha approaches $3/2$? Why did not you mention this result in the contribution part of the introduction?**
>
> We would like to clarify that the bound achieved by VR-SHSODM is better than SHSODM since the variance reduction techniques are used. Our variance reduction results do not include the case $\alpha = 3/2$. When $\alpha \rightarrow 3/2$, the bound of VR-SHSODM tends to $ \mathcal{O}(\epsilon^{-5/6})$, which is better than the $\mathcal{O}\left(\epsilon^{-4 / 3} \log (1 / \epsilon)\right)$ bound of SHSODM.
>
> **5 Do you need Lipschitzness of the gradient estimators in Theorem 4.1**
>
> We need this assumption to ensure the iteration number of the line search subroutine can be uniformly bounded (Theorem 3.1). In particular, we prove that under this assumption the desired $\delta_k$ lies in a finite interval $[\delta_l,\delta_r]$, thus the line search time can be bounded by $O(\log(1/\epsilon))$. In fact, this assumption is also used in [1, 3] when they consider solving the subproblem via gradient descent method.
>
> **6 Did you prove that the expected return in those empirical examples satisfy (alpha=1)-grad-dom.?**
>
> We do not include a complete proof in our paper, since these results have been well-established in existing papers, such as [3, 4]. They show that with the Gaussian policy, which we used in the experiments, the objective function satisfies the gradient dominance property with $\alpha=1$.
>
>
> **7 Could you please give some intuition why SHSODM outperforms TRPO and even PPO?**
>
> One of the possible explanations is our SHSODM uses the second-order information of the objective function. While PPO and TRPO both use the second-order information of the constraint (TRPO involves the constraint that the consecutive two policies should not be far away from each other, and PPO penalizes this constraint in the objective function). The information of objective function maybe more useful and can induce better policies.
>
> We again thank the reviewer for the efforts in reviewing our paper and the constructive comments!
>
>
> **References:**
>
> [1] Carmon, Y., \& Duchi, J. (2019). Gradient descent finds the cubic-regularized nonconvex Newton step. SIAM Journal on Optimization, 29(3), 2146-2178.
>
>
> [2] Cartis, C., Gould, N. I., \& Toint, P. L. (2011). Adaptive cubic regularisation methods for unconstrained optimization. Part I: motivation, convergence and numerical results. Mathematical Programming, 127(2), 245-295.
>
> [3] Masiha, S., Salehkaleybar, S., He, N., Kiyavash, N., \& Thiran, P. (2022). Stochastic second-order methods improve best-known sample complexity of sgd for gradient-dominated functions. Advances in Neural Information Processing Systems, 35, 10862-10875.
>
> [4] Yuan, R., Gower, R. M., \& Lazaric, A. (2022, May). A general sample complexity analysis of vanilla policy gradient. In International Conference on Artificial Intelligence and Statistics (pp. 3332-3380). PMLR.

---

### Official Review · Reviewer_kxnW · 2024-03-23

**Q2-1 Originality-Novelty:** 4
**Q2-2 Correctness-Technical Quality:** 3
**Q2-5 Clarity Of Writing:** 4

**Q1 Summary And Contributions:**

The paper studies the stochastic optimization of functions enjoying gradient dominance property. The authors propose the stochastic homogeneous second-order descent method, which is based on a recently proposed homogenization approach HSODM used for non-convex optimization. The authors applied two strategies to extend HSODM to gradient-dominated optimization and also enhanced their algorithm in a specific case using variance reduction techniques. Theoretically, sample complexity analyses are given, which match the best-known sample complexity in the literature. Empirically, in the context of RL, the proposed algorithm obtains great and robust performance.

**Q2-3 Extent To Which Claims Are Supported By Evidence:**

3: Good: the main claims are supported by convincing evidence (in the form of adequate experimental evaluation, proofs, (pseudo-)code, references, assumptions).

**Q2-4 Reproducibility:**

3: Good: key resources (e.g. proofs, code, data) are available and key details (e.g. proofs, experimental setup) are sufficiently well-described for competent researchers to confidently reproduce the main results.

**Q3 Main Strengths:**

1. The paper is well-written, and the main idea of the algorithm is clearly explained;
2. The method of extending HSODM to gradient-dominated optimization, along with the idea of applying variance reduction to enhance the algorithm when $\alpha \in [1, \frac{3}{2})$, are novel but easy to understand, with strong theoretical guarantee;
3. The theoretical result (sample complexity) of SHSODM is best to the current knowledge, and the actual computational cost is less than the state-of-the-art method;
4. Empirically, SHSODM produces better and more robust results compared to the current methods, and cost less in computing the update directions compared to SCRN.

**Q4 Main Weakness:**

1. The authors mention that the main difficulty of refining HSODM to gradient-dominated optimization lies in the requirement that $\lambda_k$ and $||d_k||$ should have the same order and diminish simultaneously when the algorithm proceeds. I think it'll be better if the authors can explain why this requirement matters and why the fixed strategy to choose $\delta$ works in HSODM and fails in gradient-dominated optimization;
2. SHSODM obtains best final performance in all the experiments, but as the authors mention, it grows slowly at the beginning and is inferior to the other methods in the early stage. Is there a reasonable explanation of why SHSODM grows slowly at the beginning?

**Q5 Detailed Comments To The Authors:**

Please see the Weakness above.

**Q9 Complying With Reviewing Instructions:**

Yes

---

> ### Author Rebuttal · Authors · 2024-04-05
>
> We sincerely appreciate your feedback and are grateful for the thorough reading and valuable suggestions you provided for our paper. The extensive review comments are crucial for enhancing our future revision. We hope our responses below will adequately address your concerns and questions.
>
> **1 Explain why $\lambda_k$ and $||d_k||$ should have the same order and diminish simultaneously**
>
> We can interpret this strategy using intuition from the cubic regularization algorithm. In the cubic regularization, the update rule is
> $$  x_{k+1} = x_k + d_k, \quad  d_k = \arg\min_d g_k^Td +  \frac{1}{2}d^TH_kd + C ||d||^3.$$
> The cubic term  $ C||d||^3 $ can be seen as a square term $\lambda_k||d||^2$ with 'adaptive' regularization parameter $\lambda_k = O(||d||)$. On the other hand, the update of our algorithm can be written as $x_{k+1} = x_k -  (H_k + \lambda_k I)^{-1} g_k $.  The $ \lambda_k $ in our algorithm can be viewed as the regularization parameter before the square regularization term, i.e.,  $d_k = \arg\min_d g_k^Td + \frac{1}{2}d^TH_kd + \lambda_k ||d||^2$ , and thus to choose the regularized parameter more adaptively, it should be proportional to $||d||$. This choice also facilitates the analysis.
>
> **2 Explain why the fixed strategy to choose $\delta$ works in HSODM and fails in gradient-dominated optimization**
>
> The HSODM proposed in [1] deals with a nonconvex objective function. By the optimality condition, one can safely claim that the dual variable $\theta_k \neq 0$, and the negative curvature of the augmented matrix always exists. Combined with its fixed-radius strategy, one can establish the descent lemma, and further the $\mathcal{O}(\epsilon^{-3/2})$ convergence rate. However, under the gradient-dominated setting, the objective function may be convex and the magnitude of $f-f^*$ connects directly to gradient $||g||$.
> In the original analysis, by the fixed $\delta$ we cannot connect the decrease in function value to $||g||$, and this is why a dynamic adaptation should be applied.
> This further results in an unsatisfactory rate. Therefore, we need to adaptively choose $\delta_k$ to avoid this difficulty, which motivates us to design the line search subroutine.
>
> **3 Is there a reasonable explanation of why SHSODM grows slowly at the beginning**
>
> Our algorithm requires calculating the Hessian Vector Product (HVP) using the Pearlmutter trick [2], that is,
> $$ \nabla^2 f(x)v  = \lim_{\epsilon\rightarrow 0} \frac{1}{\epsilon} (\nabla f(x+\epsilon v)-\nabla f(x)) = \nabla_x \langle \nabla f(x),v\rangle. $$ Thus, we need more function evaluations to calculate the HVP. This means that at the beginning, the first-order methods can complete more iterations than our algorithm, making first-order methods faster at the beginning. When both algorithms reach a point that is near the optimal point, the second-order methods make more progress than the first-order methods, which explains why we get a better reward at the end.
>
> We again thank the reviewer for the efforts in reviewing our paper and the constructive comments!
>
>
>
> **References:**
>
> [1] Chuwen Zhang, Dongdong Ge, Chang He, Bo Jiang, Yuntian Jiang, Chenyu Xue, and Yinyu Ye. A homogenous second-order descent method for nonconvex optimization. arXiv preprint arXiv:2211.08212, 2022.
>
> [2] Pearlmutter, B. A. (1994). Fast exact multiplication by the Hessian. Neural computation, 6(1), 147-160.

---

### Official Review · Reviewer_ygXE · 2024-03-23

**Q2-1 Originality-Novelty:** 1
**Q2-2 Correctness-Technical Quality:** 3
**Q2-5 Clarity Of Writing:** 3

**Q1 Summary And Contributions:**

The paper proposes stochastic homogeneous second-order descent method (SHSODM) for stochastic functions with gradient dominance property, based on a recently proposed homogenization approach. The paper provides sample complexity analysis for SHSODM, matching the sample complexity achieved by other second-order methods (SCRN) [Masiha et al. 2022] for gradient-dominated stochastic optimization, while gaining cheaper computational cost ($n^2$ vs $n^3$). Numerical experiments on several RL tasks demonstrate the effective performance of SHSODM.

**Q2-3 Extent To Which Claims Are Supported By Evidence:**

2: Fair: the main claims are somewhat supported by evidence (but the experimental evaluation may be weak, or does not match entirely with the claims, important baselines may be missing, proofs contain important ideas but lack rigor, algorithmic details are only discussed superficially, references are imprecise, assumptions are not sufficiently motivated or explicated, etc.).

**Q2-4 Reproducibility:**

2: Fair: key resources (e.g. proofs, code, data) are unavailable but key details (e.g. proof sketches, experimental setup) are sufficiently well-described for an expert to confidently reproduce the main results.

**Q3 Main Strengths:**

The designed algorithm (SHSODM) has the sample complexity achieved by other second-order methods (SCRN) [Masiha et al. 2022], while gaining cheaper computational cost ($n^2$ vs $n^3$).

**Q4 Main Weakness:**

While the algorithm looks promising, proposing variants of algorithm to reduce the computational cost from $n^3$ to $n^2$ may not be so importance. Indeed, for large learning problems, where $n$ is large, one would prefer to choose the first-order based algorithms, rather than the second-order based algorithms.

Also, it would be good to compare the total computational complexities and numerical performance of the proposed algorithms with state-of-the-art first-order algorithms.

**Q5 Detailed Comments To The Authors:**

Null

**Q9 Complying With Reviewing Instructions:**

Yes

---

> ### Author Rebuttal · Authors · 2024-04-05
>
> We sincerely appreciate your feedback and are grateful for the thorough reading and valuable suggestions you provided for our paper. The extensive review comments are crucial for enhancing our future revision. We hope our responses below will adequately address your concerns and questions.
>
> **1 Proposing variants of algorithm to reduce the computational cost from $n^3$ to $n^2$ may not be so important**
>
> We politely disagree with the reviewer that it is not important to reduce the computational cost from $n^3$ to $n^2$. Theoretically, improving the dependence of the computational complexity on the dimension $n$ is a critical problem in the optimization and machine learning communities. For example, although we can directly compute the inverse of $A$ to obtain the solution to the linear system $Ax=b$ with non-singular $A$ in $\mathcal{O}(n^3)$, there still are a fleet of iterative algorithms proposed to reduce the computational complexity from $\mathcal{O}(n^3)$ to $\mathcal{O}(n^2)$. Empirically, this reduction also saves expensive computational cost, and improve the efficiency.
>
> **2 On ''when $n$ is large, one would prefer to choose the first-order based algorithms, rather than the second-order based algorithms''**
>
> We respectfully disagree with the reviewer on this opinion.
>
> **Firstly**, almost all state-of-the-art first-order algorithms, such as Adam and AdaGrad, can be seen as second-order algorithms in the sense that they approximate the second-order information. Without this kind of information, the pure first-order algorithms may fail to escape the saddle point and show slow convergence in rugged landscapes of neural networks. For example, in our numerical experiments, the first-order algorithm VPG is stuck in the environment HalfCheetah-v2 and Walker2d-v2, and perform worse than our algorithm.
>
> **Secondly**, developing scalable second-order algorithms and incorporating second-order information into the algorithm design have recently become more and more popular. Even for language model pre-training, whose parameters range from 125M to 1.5B, there are some scalable second-order algorithms [2] achieving better performance and less wall-clock time than Adam.
>
> **Finally**, in practice, we do not need to store the Hessian and only need the access to Hessian-vector-product (HVP). In the implementation, we use the well-known Pearlmutter trick [3] to compute HVP, that is,
> $$ \nabla^2 f(x)v  = \lim_{\epsilon\rightarrow 0} \frac{1}{\epsilon} (\nabla f(x+\epsilon v)-\nabla f(x)) = \nabla_x \langle \nabla f(x),v\rangle. $$
> By using automated differential, the computation cost of HVP can be reduced to $\mathcal{O}(n^2)$, whose dependence on $n$ is the same as the first-order algorithms.
>
> In conclusion, we believe that this reduction is significant, and the scalable second-order algorithms are becoming more and more popular and deserve further research.
>
> **3 It would be better to compare with state-of-the-art first-order algorithms**
>
> Thanks for your suggestion. Since our algorithm is a second-order algorithm, we mainly compare it with other second-order algorithms such as SCRN and show its better performance. Besides, we also compare our algorithm with **PPO** in the appendix of this paper, **which is a widely-used first-order reinforcement learning algorithm** in practice. The experiment shows that our algorithm outperforms PPO in several environments. For other state-of-the-art first-order algorithms, such as MBPG [4], [5] already shows that SCRN performs better than them. Since our algorithm is better than SCRN empirically, it is reasonable to conclude that our algorithm also outperforms them. Finally, upon your request, we would like to compare our algorithm with some first-order algorithms that use second-order information in the revision, such as the algorithm proposed in [6].
>
> We again thank the reviewer for the efforts in reviewing our paper and the constructive comments!
>
> **References:**
>
> [1] Saad, Y. (2011). Numerical methods for large eigenvalue problems: revised edition. Society for Industrial and Applied Mathematics.
>
> [2] Liu, H., Li, Z., Hall, D. L. W., Liang, P., \& Ma, T. (2023, October). Sophia: A Scalable Stochastic Second-order Optimizer for Language Model Pre-training. In The Twelfth International Conference on Learning Representations.
>
> [3] Pearlmutter, B. A. (1994). Fast exact multiplication by the Hessian. Neural computation, 6(1), 147-160.
>
> [4] Huang, F., Gao, S., Pei, J., \& Huang, H. (2020, November). Momentum-based policy gradient methods, ICML'20.
>
> [5] Masiha, S., Salehkaleybar, S., He, N., Kiyavash, N., \& Thiran, P. (2022). Stochastic second-order methods improve best-known sample complexity of sgd for gradient-dominated functions. Advances in Neural Information Processing Systems, 35, 10862-10875.
>
> [6] Fatkhullin, I., Barakat, A., Kireeva, A., & He, N. (2023, July). Stochastic policy gradient methods: Improved sample complexity for fisher-non-degenerate policies. ICML'23.

---

### Meta-Review · Area_Chair_1L1t · 2024-04-16

The paper proposes the stochastic homogeneous second-order descent method, which is based on a recently proposed homogenization approach HSODM used for non-convex optimization. The authors have addressed the reviewers’ concerns. All the reviewers support the paper. Please add the discussions and revise the manuscript for the final version.